# Transductive Learning Is Compact

**Julian Asilis**
USC
asilis@usc.edu

**Siddartha Devic**
USC
devic@usc.edu

**Shaddin Dughmi**
USC
shaddin@usc.edu

**Vatsal Sharan**
USC
vsharan@usc.edu

**Shang-Hua Teng**
USC
shanghua@usc.edu

## Abstract

We demonstrate a compactness result holding broadly across supervised learning with a general class of loss functions: Any hypothesis class $\mathcal{H}$ is learnable with transductive sample complexity $m$ precisely when all of its finite projections are learnable with sample complexity $m$. We prove that this exact form of compactness holds for realizable and agnostic learning with respect to any *proper* metric loss function (e.g., any norm on $\mathbb{R}^d$) and any continuous loss on a compact space (e.g., cross-entropy, squared loss). For realizable learning with *improper* metric losses, we show that exact compactness of sample complexity can fail, and provide matching upper and lower bounds of a factor of 2 on the extent to which such sample complexities can differ. We conjecture that larger gaps are possible for the agnostic case. Furthermore, invoking the equivalence between sample complexities in the PAC and transductive models (up to lower order factors, in the realizable case) permits us to directly port our results to the PAC model, revealing an almost-exact form of compactness holding broadly in PAC learning.

## 1 Introduction

Compactness results in mathematics describe the behavior by which, roughly speaking, an infinite system can be entirely understood by inspecting its finite subsystems: An infinite graph is $k$-colorable precisely when its finite subgraphs are all $k$-colorable [De Bruijn and Erdös, 1951], an infinite collection of compact sets in $\mathbb{R}^d$ has non-empty intersection precisely when the same is true of its finite subcollections, etc. In each case, compactness reveals a profound and striking structure, by which local understanding of a problem immediately yields global understanding.

We demonstrate that supervised learning in the transductive model enjoys such structure. First, let us briefly review the transductive model, a close relative of the PAC model. In the realizable setting with a class of hypotheses $\mathcal{H} \subseteq \mathcal{Y}^{\mathcal{X}}$, it is defined by the following sequence of steps:

1. An adversary selects unlabeled data $S = (x_1, \ldots, x_n) \in \mathcal{X}^n$ and a hypothesis $h \in \mathcal{H}$.
2. The unlabeled datapoints $S$ are displayed to the learner.
3. One datapoint $x_i$ is selected uniformly at random from $S$. The remaining datapoints
$$S_{-i} = (x_1, \ldots, x_{i-1}, x_{i+1}, \ldots, x_n)$$
and their labels under $h$ are displayed to the learner.
4. The learner is prompted to predict the label of $x_i$, i.e., $h(x_i)$.

The expected error incurred by the learner over the uniformly random choice of $x_i$ is its *transductive error* on this learning instance, from which one can easily define the transductive sample complexity of a learner and of a hypothesis class.

38th Conference on Neural Information Processing Systems (NeurIPS 2024).

Notably, transductive learning, originally introduced by Vapnik and Chervonenkis [1974] and Vapnik [1982], is a fundamental approach to learning with deep theoretical connections to the PAC model. We study the transductive model as employed by the pioneering work of Haussler et al. [1994], who introduced the celebrated *one-inclusion graph* (OIG) to study transduction and used it to derive improved error bounds for VC classes. More recently, transductive learning and OIGs have been used to (among other work) establish the first characterizations of learnability for multiclass classification and realizable regression [Brukhim et al., 2022, Attias et al., 2023], to prove optimal PAC bounds across several learning settings [Aden-Ali et al., 2023b], and to understand regularization in multiclass learning [Asilis et al., 2024]. (See also Daniely and Shalev-Shwartz [2014], Alon et al. [2022], Montasser et al. [2022], Aden-Ali et al. [2023a].) The transductive model also naturally generalizes to the agnostic setting, much like PAC learning, as articulated by Asilis et al. [2024].

## 1.1 Contributions

Our results involve comparing a hypothesis class $\mathcal{H}$ to its "finite projections." Formally, for a hypothesis class $\mathcal{H} \subseteq \mathcal{Y}^{\mathcal{X}}$ and any finite collection of unlabeled data $S \subseteq \mathcal{X}$, we refer to the finite subsets of $\mathcal{H}|_S$ as *finite projections* of $\mathcal{H}$. Note that $\mathcal{H}$ is being "made finite" at two levels: first by restricting its functions to a finite region $S \subseteq \mathcal{X}$ of the domain, and second by passing to a finite subset of $\mathcal{H}|_S$. Thus, any finite projection of $\mathcal{H}$, e.g. $\mathcal{F} \subseteq \mathcal{H}|_S$, is necessarily a finite set of behaviors, $|\mathcal{F}| < \infty$, regardless of whether $\mathcal{H}|_S$ in its totality is infinite (as may easily occur if $\mathcal{Y}$ is infinite).

As our cornerstone result, we demonstrate in Theorem 3.6 that for the case of supervised learning with a large class of *proper*[1] metric loss functions (including any norm on $\mathbb{R}^d$ or its closed subsets; see Definition 3.2) a class $\mathcal{H}$ can be learned with transductive sample complexity $m$ precisely when the same is true of all its finite projections. In fact, in Theorem 3.7 we extend our results to arbitrary continuous losses on compact metric spaces, e.g., cross-entropy loss on finite-dimensional probability spaces and squared $\ell_2$ loss on compact subsets of $\mathbb{R}^d$. For learning over arbitrary label spaces, we demonstrate in Theorems 3.8 and 3.9 that compactness fails: for realizable learning with metric losses, we provide matching upper and lower bounds of a factor of 2 on the extent to which such transductive sample complexities can differ. Our lower bound transfers directly to transductive learning in the agnostic case, for which we conjecture that larger gaps in sample complexity are possible.

We stress that our compactness results are *exact* in the transductive model, avoiding dilution by asymptotics or even by constants. In addition, there is a growing body of work relating sample complexities in the transductive and PAC models, by which our results directly transfer in a black-box manner [Asilis et al., 2024, Aden-Ali et al., 2023b, Dughmi et al., 2024]. Notably, for realizable learning with any bounded loss, PAC sample complexities differ from their transductive counterparts by at most a logarithmic factor in $\delta$, the confidence parameter. Combined with our results, this reveals an almost-exact form of compactness for realizable PAC learning, as we describe in Section 3.4.[2]

Our results hold for improper learners, i.e., learners that are permitted to emit a predictor outside the underlying class $\mathcal{H}$. Curiously, compactness of sample complexity can be seen to fail strongly when one requires that learners be proper, using the work of Ben-David et al. [2019]. This demonstrates a structural difference between proper and improper learning; see Appendix B for further detail.

Our compactness results are underpinned by a generalization of the classic marriage theorems for bipartite graphs which may be of independent mathematical interest. The original marriage theorem, due to Philip Hall [Hall, 1935], articulates a necessary and sufficient condition for the existence of a perfect matching from one side of a *finite* bipartite graph to the other. Subsequently, Marshall Hall [Hall Jr, 1948] extended the same characterization, referencing only finite subgraphs, to infinite graphs of arbitrary cardinality, provided the side to be matched has finite degrees — the characterization being false otherwise, as can be seen by a simple countable example. This characterization therefore serves as a compactness result for matching on such infinite graphs. The proof of M. Hall features an involved analysis of the lattice of "blocking sets", and invokes the axiom of choice through Zorn's lemma. Simpler proofs have since been discovered: a topological proof by Halmos and Vaughan

---

[1]We warn that we will shortly be overloading the term "proper", as we discuss proper metric spaces and proper functions between metric spaces. We also note that our notion of properness is unrelated to losses which incentivize predicting the true probability, from e.g. Blasiok et al. [2023]. (And unrelated to proper vs. improper learners; we consider improper learners throughout the paper, which can emit predictors outside the class $\mathcal{H}$.)

[2]Note too that any future improvements to the connections between the PAC and transductive models, whether in the realizable or agnostic settings, will be automatically inherited by our results in a black-box manner.

[1950] which invokes the axiom of choice through Tychonoff's theorem, and an algebraic proof by Rado [1967] which also uses Zorn's lemma. At the heart of our paper is a compactness result (Theorem 3.3) for a variable-assignment problem which generalizes both supervised learning and bipartite matching: one side of a bipartite graph indexes infinitely many variables, the other indexes infinitely many functions that depend on finitely many variables each, and the goal is to assign all the variables in a manner that maintains all functions below a target value. Our proof draws inspiration from all three of the aforementioned proofs of M. Hall's theorem, and goes through Zorn's lemma.

## 1.2 Related Work

The transductive approach to learning dates to the work of Vapnik and Chervonenkis [1974] and Vapnik [1982], and has inspired a breadth of recent advances across regression, classification, and various other learning regimes; see our introduction for a brief overview. Regarding transductive sample complexities, Hanneke et al. [2023] recently demonstrated a trichotomy result for optimal transductive error rates in the *online* setting of Ben-David et al. [1997]. In contrast, we focus on the classical (batch) setting, as described in Section 2.2.

Perhaps most related to the present work is Attias et al. [2023], which introduces the $\gamma$-OIG dimension and demonstrates that it characterizes learnability for supervised learning problems with pseudometric losses. Notably, this is the first general dimension characterizing learnability across essentially the entirety of supervised learning. The $\gamma$-OIG dimension itself establishes a qualitative form of compactness — as it is defined using only the finite projections of a class — but we note that it has not been shown to tightly characterize the sample complexity of learning. Furthermore, it is analyzed only for realizable learning, which is in general not equivalent to agnostic learning (e.g., for regression). Our work, in contrast, establishes exact compactness for the sample complexity of transductive learning for both the realizable and agnostic settings, with respect to a general class of loss functions. Moreover, in Appendix B we extend our results to certain cases of distribution-family learning, including realizable learning of partial concept classes.

# 2 Preliminaries

## 2.1 Notation

For a natural number $n \in \mathbb{N}$, $[n]$ denotes the set $\{1, \ldots, n\}$. For a predicate $P$, $[P]$ denotes the Iverson bracket of $P$, i.e., $[P] = 1$ when $P$ is true and 0 otherwise. When $Z$ is a set, $Z^{<\omega}$ denotes the set of all finite sequences in $Z$, i.e., $Z^{<\omega} = \bigcup_{i=1}^{\infty} Z^i$. For a tuple $S = (z_1, \ldots, z_n)$, we use $S_{-i}$ to denote $S$ with its $i$th entry removed, i.e., $S_{-i} = (z_1, \ldots, z_{i-1}, z_{i+1}, \ldots, z_n)$.

## 2.2 Transductive Learning

Let us recall the standard toolkit of supervised learning. A learning problem is determined by a **domain** $\mathcal{X}$, **label space** $\mathcal{Y}$, and **hypothesis class** $\mathcal{H} \subseteq \mathcal{Y}^{\mathcal{X}}$. The elements of $\mathcal{H}$ are functions $\mathcal{X} \to \mathcal{Y}$; such functions are referred to as **hypotheses** or **predictors**. Learning also requires a **loss function** $\ell$ (or $d$) from $\mathcal{Y} \times \mathcal{Y}$ to $\mathbb{R}_{\geq 0}$, which often endows $\mathcal{Y}$ with the structure of a metric space. Throughout the paper, we permit $\mathcal{X}$ to be arbitrary. A **labeled datapoint** is a pair $(x, y) \in \mathcal{X} \times \mathcal{Y}$ and an **unlabeled datapoint** is an element $x \in \mathcal{X}$. A **training set**, or *training sample*, is a tuple of labeled datapoints $S \in (\mathcal{X} \times \mathcal{Y})^{<\omega}$. A **learner** is a function from training sets to predictors, i.e., $A : (\mathcal{X} \times \mathcal{Y})^{<\omega} \to \mathcal{Y}^{\mathcal{X}}$.

**Definition 2.1.** *Realizable **transductive learning** is defined as follows: An adversary selects $S = (x_i)_{i \in [n]} \in \mathcal{X}^n$ and a hypothesis $h \in \mathcal{H}$. The unlabeled datapoints $S$ are displayed to the learner. Then one datapoint $x_i$ is selected uniformly at random from $S$, and the remaining datapoints and their labels under $h$ are displayed to the learner. Lastly, the learner is prompted to predict the label of $x_i$, i.e., $h(x_i)$.*

We refer to the information of $(S, h)$ as in Definition 2.1 as an **instance** of transductive learning, and to $x_i$ as the (randomly selected) **test datapoint** and $S_{-i}$ the (randomly selected) **training datapoints**. The **transductive error** incurred by a learner $A$ on an instance $(S, h)$ is its average error over the uniformly random choice of test datapoint, i.e.,

$$L_{S,h}^{\text{Trans}}(A) = \frac{1}{n} \sum_{i \in [n]} \ell\big(A(S_{-i}, h)(x_i), h(x_i)\big),$$

where $A(S_{-i}, h)$ denotes the output of $A$ on the sample $(x_j, h(x_j))_{x_j \in S_{-i}}$.

Having defined transductive error, it is natural to define error rates and sample complexity.

**Definition 2.2.** *The **transductive error rate** of a learner $A$ for $\mathcal{H}$ is the function $\xi_{A,\mathcal{H}} : \mathbb{N} \to \mathbb{R}$ defined by $\xi_{A,\mathcal{H}}(n) = \sup_{S \in \mathcal{X}^n, h \in \mathcal{H}} L_{S,h}^{\mathrm{Trans}}(A)$. The **transductive sample complexity** of a learner $A$ for $\mathcal{H}$ is the function $m_{\mathrm{Trans}, A}(\epsilon) = \min\{m \in \mathbb{N} : \xi_{A,\mathcal{H}}(m') \leq \epsilon, \forall m' \geq m\}$.*

**Definition 2.3.** *The **transductive error rate** of a class $\mathcal{H}$ is the minimal error rate attained by any of its learners, i.e., $\xi_{\mathcal{H}}(n) = \inf_A \xi_{A,\mathcal{H}}(n)$. The **transductive sample complexity** $m_{\mathrm{Trans}, \mathcal{H}} : \mathbb{R}_{>0} \to \mathbb{N}$ of $\mathcal{H}$ is the function mapping $\epsilon$ to the minimal $m$ for which $\xi_{\mathcal{H}}(m') \leq \epsilon$ for all $m' \geq m$. That is,*

$$m_{\mathrm{Trans}, \mathcal{H}}(\epsilon) = \min\{m \in \mathbb{N} : \xi_{\mathcal{H}}(m') \leq \epsilon, \forall m' \geq m\}.$$

*We say that $\mathcal{H}$ is learnable in the realizable case with transductive sample function $m$ when $m_{\mathrm{Trans}, \mathcal{H}}(\epsilon) \leq m(\epsilon)$ for all $\epsilon$.*

Informally, agnostic transductive learning is the analogue in which the adversary is permitted to label the data in $S$ arbitrarily, and in which the learner need only compete with the best hypothesis in $\mathcal{H}$. We defer the formal definition to Section 3.3.

## 3   Compactness of Learning

We present the central result of the paper in this section: the transductive sample complexity of learning is a compact property of a hypothesis class. In Section 3.1 we study compactness of realizable supervised learning over *proper* loss functions, and demonstrate a strong compactness result: a class $\mathcal{H}$ is learnable with transductive sample complexity $m$ if and only if all its finite projections are learnable with the same complexity. In Section 3.2 we examine the case of realizable supervised learning over improper loss functions and prove a negative result: the previous compactness result no longer holds in this more general setting. Nevertheless, we demonstrate an approximate form of compactness, up to a factor of 2, for (improper) metric losses. Moreover, we show exact compactness for the special case of the (improper) 0-1 loss function, i.e., multiclass classification over arbitrary, possibly infinite label sets. Notably, this recovers M. Hall's classic matching theorem for infinite graphs [Hall Jr, 1948] as a corollary to our central result. In Section 3.3 we examine analogues of our results for agnostic learning, and in Section 3.4 we transfer our results to the PAC model via standard equivalences, obtaining approximate compactness of sample complexities. Due to space constraints, we defer an extension of our results to distribution-family PAC learning to Appendix B.

### 3.1   Realizable Learning With Proper Loss Functions

We first consider the case of loss functions $\ell : \mathcal{Y} \times \mathcal{Y} \to \mathbb{R}_{\geq 0}$ defined on a proper metric space $\mathcal{Y}$.

**Definition 3.1.** *A metric space is **proper** if its closed and bounded subsets are all compact.*

A related notion is that of a proper map between metric spaces.

**Definition 3.2.** *A function $f : X \to Y$ between metric spaces is **proper** if it reflects compact sets, i.e., $f^{-1}(U) \subseteq X$ is compact when $U \subseteq Y$ is compact.*

We remark that proper spaces are sometimes referred to as *Heine-Borel spaces*, and that their examples include $\mathbb{R}^d$ endowed with any norm, all closed subsets of $\mathbb{R}^d$ (under the same norms), and all finite sets endowed with arbitrary metrics. Further discussion of proper metric spaces is provided in Appendix A. The central technical result of this subsection is a compactness property concerning assignments of variables to metric spaces that maintain a family of functions below a target value $\epsilon$.

**Theorem 3.3.** *Let $L$ be a collection of variables, with each variable $\ell \in L$ taking values in a metric space $M_\ell$. Let $R$ be a collection of proper functions, each of which depends upon finitely many variables in $L$ and has codomain $\mathbb{R}_{\geq 0}$. Then the following conditions are equivalent for any $\epsilon > 0$.*

1. *There exists an assignment of all variables in $L$ which keeps the output of each function $r \in R$ no greater than $\epsilon$.*

2. *For each finite subset $R'$ of $R$, there exists an assignment of all variables in $L$ which keeps the output of each function $r' \in R'$ no greater than $\epsilon$.*

*Proof.* (1.) $\implies$ (2.) is immediate. Before arguing the reverse direction, some terminology: a *partial assignment* of variables is an assignment of variables for a subset of $L$. A partial assignment is said to be *completable* with respect to $R' \subseteq R$ if its unassigned variables can all be assigned so that all functions $r' \in R'$ are kept below $\epsilon$. A partial assignment is *finitely completable* if it is completable with respect to all finite subsets of $R$. This is a pointwise condition: the completions are permitted to vary across $R$'s subsets.

> **Lemma 3.4.** *Given a finitely completable partial assignment with an unassigned variable, one such variable can be assigned while preserving finite completability.*
>
> *Proof.* Fix any unassigned variable $\ell \in L$; we will assign it while preserving finite completability. For each set $R' \subseteq R$, let $N_{R'}(\ell)$ consist of those assignments of $\ell$ that preserve completability with respect to $R'$. By the assumption of finite completability, we have that $N_{R'}(\ell)$ is non-empty for all finite $R'$. We claim furthermore that $N_{R'}(\ell)$ is compact for finite $R'$.
>
> To see why, let $|R'| = k$ and let $\ell_1, \ldots, \ell_m$ be the variables in $L$ upon which the functions in $R'$ depend. Suppose without loss of generality that $\ell = \ell_1$ and that nodes $\ell_{i+1}, \ldots, \ell_m$ have already been assigned. Consider the function $f_{R'} : \prod_{j=1}^{i} M_{\ell_j} \to \mathbb{R}^k$ mapping assignments of the $\ell_1, \ldots, \ell_i$ to the outputs they induce on the functions in $R'$. (Notably, this includes the assignments already made for $\ell_{i+1}, \ldots, \ell_m$.) As the functions in $R'$ are proper, including when fixing some of their inputs, $f_{R'}$ is as well.
>
> Thus $f_{R'}^{-1}([0, \epsilon]^k)$ is compact, as is its projection onto its first coordinate. That set is precisely $N_{R'}(\ell)$, demonstrating our intermediate claim. We thus have a family of compact, non-empty sets $\mathcal{I} = \{N_{R'}(\ell) : R' \subseteq R, |R'| < \infty\}$. Note that finite intersections of elements of $\mathcal{I}$ are non-empty, as $\bigcap_{i=1}^{j} N_{R_i}(\ell) \supseteq N_{\bigcup_{i=1}^{j} R_i}(\ell) \neq \emptyset$.
>
> In metric spaces, an infinite family of compact sets has non-empty intersection if and only if the same holds for its finite intersections. Thus, by compactness of each element of $\mathcal{I}$, the intersection across all of $\mathcal{I}$ is non-empty. That is, there exists an assignment for $\ell$ which is completable with respect to all finite subsets of $R$. The claim follows. $\square$

We now complete the argument using Zorn's lemma. Let $\mathcal{P}$ be the poset whose elements are finitely completable assignments, where $\phi_1 \leq \phi_2$ if $\phi_2$ agrees with all assignments made by $\phi_1$ and perhaps assigns additional variables. Note first that chains in $\mathcal{P}$ have upper bounds. In particular, let $\mathcal{C} \subseteq \mathcal{P}$ be a chain and define $\phi_\mathcal{C}$ to be the "union" of assignments in $\mathcal{C}$, i.e., $\phi_\mathcal{C}$ leaves $\ell$ unassigned if all $\phi \in \mathcal{C}$ leave $\ell$ unassigned, otherwise assigns $\ell$ to the unique element used by assignments in $\mathcal{C}$.

Clearly $\phi_\mathcal{C}$ serves as an upper bound of $\mathcal{C}$, provided that $\phi_\mathcal{C} \in \mathcal{P}$. To see that $\phi_\mathcal{C} \in \mathcal{P}$, fix a finite set $R' \subseteq R$. $R'$ is incident to a finite collection of nodes in $L$, say $\ell_1, \ldots, \ell_m$. Suppose $\ell_1, \ldots, \ell_i$ are those which are assigned by $\phi_\mathcal{C}$, and let $\phi_1, \ldots, \phi_i \in \mathcal{C}$ be assignments which assign (i.e., do *not* leave free) the respective nodes $\ell_1, \ldots, \ell_i$. Then, as $\mathcal{C}$ is totally ordered, it must be that one of $\phi_1, \ldots, \phi_i$ assigns all of the variables $\ell_1, \ldots, \ell_i$. That is, there exists $\phi_j \in \mathcal{C}$ which agrees with $\phi_\mathcal{C}$ in its action on $\ell_1, \ldots, \ell_m$. As $\phi_j \in \mathcal{P}$, it must be that $\phi_j$ is completable with respect to $R'$. Then $\phi_\mathcal{C}$ is also completable with respect to $R'$, as $R'$ depends only upon $\ell_1, \ldots, \ell_m$.

Thus, invoking Zorn's lemma, $\mathcal{P}$ has a maximal element $\phi_{\max}$. By Lemma 3.4, it must be that $\phi_{\max}$ does not leave a single variable unassigned, otherwise it could be augmented with an additional assignment. There thus exists a *total* assignment that is finitely completable. As it has no free variables, it must indeed be maintaining all functions in $R$ below $\epsilon$. The claim follows. $\square$

**Remark 3.5.** *A corollary to Theorem 3.3 is that the same claim holds when the target values $\epsilon$ vary over the functions $r \in R$, as translations and scalings of proper functions $Z \to \mathbb{R}$ are proper.*

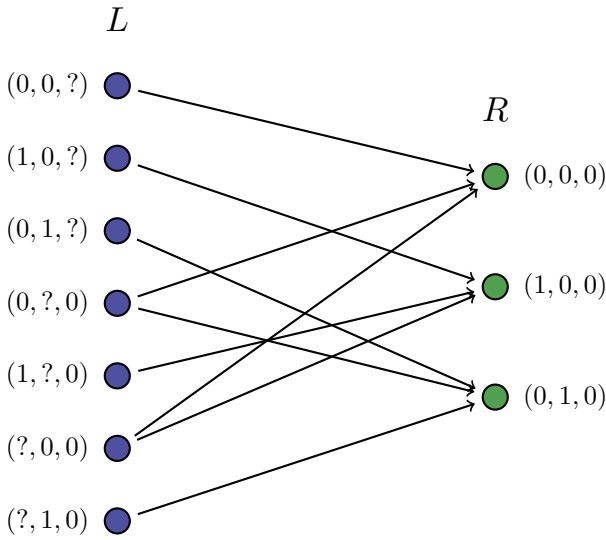

Figure 1: Depiction of variables $L$ and functions $R$ which model transductive learning, for a sequence of unlabeled datapoints $|S| = 3$ such that $\mathcal{H}|_S$ contains the behaviors $(0, 0, 0)$, $(1, 0, 0)$, and $(0, 1, 0)$. Arrows denote functional dependence, i.e., each $r \in R$ depends upon its incident variables.

**Theorem 3.6.** *Let $\mathcal{X}$ be an arbitrary domain, $\mathcal{Y}$ a label set, and $d$ a loss function such that $(\mathcal{Y}, d)$ is a proper metric space. Then the following are equivalent for any $\mathcal{H} \subseteq \mathcal{Y}^{\mathcal{X}}$ and $m \colon \mathbb{R}_{>0} \to \mathbb{N}$:*

1. *$\mathcal{H}$ is learnable in the realizable case with transductive sample function $m$.*

2. *For any finite $X \subseteq \mathcal{X}$ and finite $\mathcal{H}' \subseteq \mathcal{H}|_X$, $\mathcal{H}'$ is learnable in the realizable case with transductive sample function $m$.*

*Proof.* (1.) $\implies$ (2.) is immediate. For the reverse direction, fix an $\epsilon > 0$ and set $n = m(\epsilon)$. Then fix a sequence of unlabeled datapoints $S \in \mathcal{X}^n$. It suffices to demonstrate that a transductive learner for $\mathcal{H}$ on instances of the form $\{(S, h)\}_{h \in \mathcal{H}}$ can be designed which attains error $\le \epsilon$.

We will capture the task of transductively learning $\mathcal{H}$ on such instances by way of a certain collection $R$ of functions and $L$ of variables. Each variable $\ell \in L$ will be permitted to take values in $\mathcal{Y}$, while each function $r \in R$ depends upon exactly $n$ variables in $L$ and outputs values in $\mathbb{R}_{\ge 0}$. More precisely, let $R = \mathcal{H}|_S$ and $L = \bigcup_{S' \subseteq S, |S'| = n-1} \mathcal{H}|_{S'}$. These serve merely as representations for the functions in $R$ and variables in $L$, not their true definitions (which will be established shortly). Note now that by suppressing the unlabeled datapoints of $S$, we can equivalently represent elements of $R$ as sequences in $\mathcal{Y}^n$ and elements of $L$ as sequences in $(\mathcal{Y} \cup \{?\})^n$. In this view, each element of $L$ is precisely an element of $R$ which had exactly one entry replaced with a "?". See Figure 1.

Now, to model transductive learning, fix an element $r \in R$ represented by $(y_1, \dots, y_n) \in \mathcal{Y}^n$. Then we will define $r$ to be a function depending upon the variables $\ell_1, \dots, \ell_n \in L$, where $\ell_i = (y_1, \dots, y_{i-1}, ?, y_{i+1}, \dots, y_n)$. Given assignments for each of the variables $\ell_1, \dots, \ell_n$ as values in $\mathcal{Y}$ — semantically, completions of their "?" entries — the node $r$ then outputs the value $\frac{1}{n} \cdot \sum_{i=1}^{n} d(y_i, \ell_i)$. The two crucial observations are as follows: an assignment of each $\ell \in L$ corresponds precisely to the action of a learner responding to a query at test time, and the output of node $r$ equals the error of a learner when $r$ is the ground truth.

Thus, it remains to show that the variables in $L$ can all be assigned so as to keep the outputs of the functions in $R$ less than $\epsilon$. The condition (2.) grants us that this is true for each finite collection of functions $R' \subseteq R$. Now note that the functions $r \in R$ are proper, as each such $r$ is continuous and reflects bounded sets, and as $\mathcal{Y}$ itself is proper. Invoke Theorem 3.3 to complete the proof. $\qquad \square$

Theorem 3.6 establishes an exact compactness in learning with respect to a flexible class of metric loss functions. One may note, however, that some non-metric losses are of central importance to

machine learning, including the squared error on compact subsets of $\mathbb{R}$ (which violates the triangle inequality) and the cross-entropy loss for finite-dimensional distributions (which is not symmetric). We now provide a modified form of Theorem 3.6 which captures these loss functions, in which the loss function $\ell_{\mathcal{Y}}$ is permitted to differ from the underlying metric $d$ on $\mathcal{Y}$. (E.g., such that $d$ is the usual Euclidean norm on a compact subset $\mathcal{Y}$ of $\mathbb{R}^d$, and $\ell_{\mathcal{Y}}$ is any continuous loss function.)

**Theorem 3.7.** *Let $\mathcal{X}$ be an arbitrary domain, $(\mathcal{Y}, d)$ a compact metric space, and $\mathcal{H} \subseteq \mathcal{Y}^{\mathcal{X}}$ a hypothesis class. Let $\ell_{\mathcal{Y}} : \mathcal{Y} \times \mathcal{Y} \to \mathbb{R}_{\geq 0}$ be a loss function employed for learning that is continuous with respect to the metric $d$. Then the following are equivalent for any $m \colon \mathbb{R}_{>0} \to \mathbb{N}$:*

1. *$\mathcal{H}$ is learnable in the realizable case with transductive sample function $m$.*

2. *For any finite $X \subseteq \mathcal{X}$ and finite $\mathcal{H}' \subseteq \mathcal{H}|_X$, $\mathcal{H}'$ is learnable in the realizable case with transductive sample function $m$.*

*Proof.* We adopt precisely the perspective of Theorem 3.6, seeing transductive learning modeled as a variable assignment problem with the same variables $L$ and functions $R$. To invoke Theorem 3.3, it remains only to show that the functions $r \in R$ are proper. First note a continuous function from a compact space to $\mathbb{R}$ is automatically proper, as closed subsets of compact sets are compact. Now recall that each $r \in R$ is a sum of scaled copies of $\ell_{\mathcal{Y}}$ with one input fixed. As each such function is continuous, $r$ itself is continuous and thus proper. $\square$

## 3.2 Realizable Learning With Improper Loss Functions

It is natural to ask whether the requirement that $\mathcal{Y}$ be a proper metric space is essential to Theorem 3.6 or merely an artifact of the proof. We now demonstrate the former: for arbitrary metric losses, the error rate of learning $\mathcal{H}$ can exceed that of all its finite projections by a factor of 2. Recall that $\xi_{\mathcal{H}} : \mathbb{N} \to \mathbb{R}_{\geq 0}$ denotes the transductive error rate of learning a class $\mathcal{H}$, i.e., $\xi_{\mathcal{H}}(n)$ denotes the error incurred by an optimal learner for $\mathcal{H}$ on (worst-case) samples of size $n$.

**Theorem 3.8.** *There exists a hypothesis class $\mathcal{H} \subseteq \mathcal{Y}^{\mathcal{X}}$, metric loss function $d$ on $\mathcal{Y}$, and $n \in \mathbb{N}$ such that for any finite $X \subseteq \mathcal{X}$ and finite $\mathcal{H}' \subseteq \mathcal{H}|_X$, $\xi_{\mathcal{H}}(n) \geq 2 \cdot \xi_{\mathcal{H}'}(n)$.*

Let us describe the main idea of Theorem 3.8, whose proof is deferred to Appendix C.1. The crucial step lies in the creation of the label space $\mathcal{Y} = R \cup S$, where $R$ is an infinite set whose points are all distance 2 apart, and $S$ is an infinite set whose elements are indexed by the finite subsets of $R$, e.g., as in $s_{R'} \in S$ for finite $R' \subseteq R$. For all such $R'$, define $s_{R'}$ to be distance 1 from the elements of $R'$, distance 2 from the other elements of $R$, and distance 1 from all other points in $S$. Then $\mathcal{Y}$ indeed forms a metric space, and it is straightforward to see that, for instance, the class of all functions from a one-element set to $\mathcal{Y}$ is more difficult to learn than its finite projections (equivalently, finite subsets).

We now prove a matching upper bound to Theorem 3.8, demonstrating that a factor of 2 is the greatest possible gap between the error rate of $\mathcal{H}$ and its projections when the loss function is a metric.

**Theorem 3.9.** *Let $\mathcal{Y}$ be a label set with a metric loss function and $\mathcal{H} \subseteq \mathcal{Y}^{\mathcal{X}}$ a hypothesis class. Fix $\xi : \mathbb{N} \to \mathbb{R}_{\geq 0}$, and suppose that for any finite $X \subseteq \mathcal{X}$ and finite $\mathcal{H}' \subseteq \mathcal{H}|_X$, $\mathcal{H}'$ has transductive error rate $\xi_{\mathcal{H}'} \leq \xi$. Then $\mathcal{H}$ has transductive error rate $\xi_{\mathcal{H}} \leq 2 \cdot \xi$.*

The proof of Theorem 3.9 is deferred to Appendix C.2, but let us briefly sketch the main idea. Fix $n \in \mathbb{N}$, $S \in \mathcal{X}^n$, and set $\epsilon = \xi(n)$. Consider again the collection of functions $R = \mathcal{H}|_S$ and variables $L = \bigcup_{S' \subseteq S, |S'|=n-1} \mathcal{H}|_{S'}$, as described in the proof of Theorem 3.6. By the premise of the theorem, for any finite subset $R' \subseteq R$, there exists an assignment of variables $L \to \mathcal{Y}$ which maintains all functions in $R'$ below $\epsilon$. Each such assignment induces an *apportionment* of error to each function in $R'$, i.e., a vector of length $n$ with positive entries summing to $\epsilon$. For $r \in R'$ depending upon variables $\ell_1, \ldots, \ell_n$, this apportionment tracks the contribution of each $\ell_i$ to the output of $r$. The central technical step of the proof is to demonstrate that one can assign apportionments to each node $r \in R$ such that any finite subset of the apportionments can be satisfied by an assignment of variables $L \to \mathcal{Y}$. Then let $\ell = (y_1, \ldots, y_{i-1}, ?, y_{i+1}, \ldots, y_n)$ be a variable. We assign $\ell$ to the value $\hat{y}$ such that the function $(y_1, \ldots, y_i, \hat{y}, y_{i+1}, \ldots, y_n)$ has minimal budget apportioned to $\ell$, among all such $\hat{y}$. From an invocation of the triangle inequality, this learner at most doubles the output of any $r \in R$.

Recall from Section 3.1 that proper metric spaces are sufficiently expressive to describe many of the most frequently studied label spaces, including $\mathbb{R}^d$ (equipped with any norm) and its closed subsets.

What, then, is a typical example of a label space which fails to be proper? Perhaps the most natural example is multiclass classification over infinite label sets, i.e., $\mathcal{Y}$ equipped with the discrete metric $\ell_{0-1}(y, y') = [y \neq y']$. We will now demonstrate, however, that the particular structure of multiclass classification can be exploited to recover an exact compactness result in the style of Theorem 3.6. Notably, we do so by invoking M. Hall's classic matching theorem for infinite graphs, which for good measure we show to be a special case of our Theorem 3.3.

**Definition 3.10.** *Let $G = (L \cup R, E)$ be a bipartite graph. An $R$-**matching** is a set $E' \subseteq E$ of disjoint edges which covers $R$. A graph with an $R$-matching is said to be $R$-**matchable**.*

**Definition 3.11.** *A bipartite graph $G = (L \cup R, E)$ is **finitely $R$-matchable** if for each finite subset $R'$ of $R$, there exists a set $E' \subseteq E$ of disjoint edges which covers $R'$.*

M. Hall's theorem states that an infinite bipartite graph $G$ is $R$-matchable if and only if it is finitely $R$-matchable, provided that all nodes in $R$ have finite degree. Before proving M. Hall's theorem by way of Theorem 3.3, we establish an intermediate lemma.

**Lemma 3.12.** *Let $G = (L \cup R, E)$ be a bipartite graph such that all nodes $r \in R$ have finite degree and $G$ is finitely $R$-matchable. Then there exists a collection of edges $E' \subseteq E$ such that $G' = (L \cup R, E')$ is finitely $R$-matchable and all nodes in $G'$ have finite degree.*

The proof of Lemma 3.12 is deferred to Appendix C.3, but its intuition is fairly simple: by P. Hall's theorem, $G$ is finitely $R$-matchable precisely when Hall's condition holds, i.e., $|N(R')| \geq |R'|$ for all finite $R' \subseteq R$ [Hall, 1935]. Thus any $\ell \in L$ which is not incident to a Hall blocking set can be removed from $G$ while preserving Hall's condition and finite $R$-matchability. Proceeding in this way, nodes can be removed until each remaining $\ell \in L$ is contained in a Hall blocking set $R'_\ell$. At this point, $\ell$'s incident edges can be safely restricted to those which are incident with $R'_\ell$, a finite set.

We now prove M. Hall's theorem as a consequence of our Theorem 3.3.

**Theorem 3.13** (Hall Jr [1948])**.** *Let $G = (L \cup R, E)$ be a bipartite graph in which all nodes $r \in R$ have finite degree. Then $G$ has an $R$-matching if and only if it is finitely $R$-matchable.*

*Proof.* The forward direction is clear. For the reverse, suppose $G$ is finitely $R$-matchable. Then we may assume as a consequence of Lemma 3.12 that the nodes in $L$ have finite degree as well. Let us think of each node $\ell \in L$ as a variable residing in the discrete metric space on its neighbors. We will also think of each node $r \in R$ as a function of its neighbors, which outputs the number of neighbors that have not been assigned to $r$ itself. Note that the discrete metric space on finitely many elements is proper, and furthermore that any function out of such a space is automatically proper. Then invoke Theorem 3.3 with $\epsilon(r) = 1 - \frac{1}{\deg(r)}$ to complete the proof. (See Remark 3.5.) $\qquad \square$

**Corollary 3.14.** *Let $\mathcal{H} \subseteq \mathcal{Y}^{\mathcal{X}}$ be a classification problem, i.e., employing the 0-1 loss function. Then the following are equivalent for any $m \colon \mathbb{R}_{>0} \to \mathbb{N}$:*

1. *$\mathcal{H}$ is learnable in the realizable case with transductive sample function $m$.*

2. *For any finite $X \subseteq \mathcal{X}$ and finite $\mathcal{H}' \subseteq \mathcal{H}|_X$, $\mathcal{H}'$ is learnable in the realizable case with transductive sample function $m$.*

*Proof.* Certainly (1.) $\implies$ (2.). Then suppose (2.) and fix $S \in \mathcal{X}^n$. Now consider the bipartite graph $G = (L \cup R, E)$ with $R = \mathcal{H}|_S$, $L = \bigcup_{S' \subseteq S, |S'|=n-1} \mathcal{H}|_{S'}$, and where edges in $E$ connect functions agreeing on common inputs. Then a learner for instances of the form $\{(S, h)\}_{h \in \mathcal{H}}$ amounts precisely to a choice of incident node (equivalently, edge) for each $\ell \in L$. Furthermore, such a learner attains error $\leq \epsilon$ precisely when its selected edges contribute indegree at least $d = n \cdot (1 - \epsilon)$ to each node in $R$. Using a splitting argument (i.e., creating $d$ copies of each node in $R$), this is equivalent to asking for an $R$-perfect matching in a graph which, by (2.), is finitely $R$-matchable. Note that each node in $R$ has degree $n < \infty$ and appeal to Theorem 3.13 to complete the proof. $\qquad \square$

### 3.3 Agnostic Learning

Our discussion thus far has restricted attention to realizable learning: what can be said of the agnostic case? In short, all results from Sections 3.1 and 3.2 can be claimed for agnostic learning (with nearly identical proofs), with the exception of Theorem 3.9. To begin, let us briefly review transductive learning in the agnostic case. See Asilis et al. [2024] or Dughmi et al. [2024] for further detail.

**Definition 3.15.** *The setting of **transductive learning in the agnostic case** is defined as follows:*

1. *An adversary selects a collection of $n$ labeled datapoints $S \in (\mathcal{X} \times \mathcal{Y})^{<\omega}$.*

2. *The unlabeled datapoints in $S$ are all revealed to the learner.*

3. *One labeled datapoint $(x_i, y_i)$ is selected uniformly at random from $S$. The remaining labeled datapoints $S_{-i}$ are displayed to the learner.*

4. *The learner is prompted to predict the label of $x_i$.*

Notably, transductive learning in the agnostic case differs from the realizable case in that the adversary is no longer restricted to label the datapoints in $S$ using a hypothesis $\mathcal{H}$. To compensate for the increased difficulty, and in accordance with the PAC definition of agnostic learning, a learner is only judged relative to best-in-class performance across $\mathcal{H}$. Formally,

$$L_S^{\mathrm{Trans}}(A) = \frac{1}{n} \sum_{i \in [n]} \ell(A(S_{-i})(x_i), y_i) - \inf_{h \in \mathcal{H}} \frac{1}{n} \sum_{i \in [n]} \ell(h(x_i), y_i).$$

Furthermore, one can use nearly identical reasoning as in the proofs of Theorems 3.6 and 3.7 to see that agnostic transductive learning is described by a system of variables $L$ and functions $R$. In particular, set $R = \mathcal{Y}^n$ and let $L \subseteq (\mathcal{Y} \cup \{?\})^n$ contain all sequences with exactly one "?". Then a function $r = (y_1, \ldots, y_n)$ depends upon the variables $\{\ell_i\}_{i \in [n]}$, where $\ell_i = (y_1, \ldots, y_{i-1}, ?, y_{i+1}, \ldots, y_n)$ and $r(\ell_1, \ldots, \ell_n) = \frac{1}{n} \sum_{i \in [n]} d_{\mathcal{Y}}(y_i, \ell_i) - \inf_{h \in \mathcal{H}} \frac{1}{n} \sum_{i \in [n]} \ell(h(x_i), y_i)$.

Now, as in the realizable case, a learner $A$ corresponds precisely to an assignment of each variable $\ell \in L$ to a value in $\mathcal{Y}$, and $A$ incurs agnostic transductive error at most $\epsilon$ if and only if the outputs of all nodes in $R$ are maintained below $\epsilon$. Under the conditions of Theorems 3.6 or 3.7, exact compactness of sample complexity thus comes as an immediate consequence of Theorem 3.3 and our preceding discussion. Furthermore, when $\mathcal{Y}$ bears a discrete metric, learning reduces to an assignment problem in graphs, and exact compactness follows from a straightforward splitting argument applied to M. Hall's matching theorem (as in Corollary 3.14). We thus have the following theorem.

**Theorem 3.16.** *Let $\mathcal{Y}$ and the loss function satisfy the conditions of Theorem 3.6, Theorem 3.7, or Corollary 3.14. Then the following conditions are equivalent for any $\mathcal{H} \subseteq \mathcal{Y}^{\mathcal{X}}$ and $m \colon \mathbb{R}_{>0} \to \mathbb{N}$:*

1. *$\mathcal{H}$ is learnable in the agnostic case with transductive sample function $m$.*

2. *For any finite $X \subseteq \mathcal{X}$ and finite $\mathcal{H}' \subseteq \mathcal{H}|_X$, $\mathcal{H}'$ is learnable in the agnostic case with transductive sample function $m$.*

Regarding improper metric losses, note that our lower bound from Theorem 3.8 transfers directly to the agnostic case, as it established for a hypothesis class for which agnostic learning is precisely as difficult as realizable learning. We conjecture that larger differences in such error rates — perhaps of arbitrarily large ratio — are possible for the agnostic case.

### 3.4 PAC Learning

Though our results have thus far been phrased in the language of transductive learning, we now demonstrate that they may be easily extended (in an approximate manner) to Valiant's celebrated PAC model [Valiant, 1984]. The PAC model makes use of probability measures $D$ over $\mathcal{X} \times \mathcal{Y}$, for which the *true error* incurred by a predictor $h$ is defined as $L_D(h) = \mathbb{E}_{(x,y) \sim D} \ell(h(x), y)$.

**Definition 3.17.** *Let $\mathbb{D}$ be a collection of probability measures over $\mathcal{X} \times \mathcal{Y}$ and $\mathcal{H} \subseteq \mathcal{Y}^{\mathcal{X}}$ a hypothesis class. A learner $A$ is a **PAC learner for $\mathcal{H}$ with respect to** $\mathbb{D}$ if there exists a **sample function** $m \colon (0,1)^2 \to \mathbb{N}$ such that the following holds: for any $D \in \mathbb{D}$ and $\epsilon, \delta \in (0,1)^2$, a $D$-i.i.d. sample $S$ with $|S| \geq m(\epsilon, \delta)$ is such that, with probability at least $1 - \delta$ over the choice of $S$,*

$$L_D(A(S)) \leq \inf_{\mathcal{H}} L_D(h) + \epsilon.$$

*Agnostic PAC learning* refers to the case in which $\mathbb{D}$ consists of all measures over $\mathcal{X} \times \mathcal{Y}$, and *realizable PAC learning* to the case in which $\mathbb{D} = \{D : \min_{\mathcal{H}} L_D(h) = 0\}$.

**Definition 3.18.** *The **sample complexity** of a learner $A$ with respect to a hypothesis class $\mathcal{H}$, $m_{\mathrm{PAC},A} : (0,1)^2 \to \mathbb{N}$, is the minimal sample function it attains as a learner for $\mathcal{H}$. The **sample complexity** of a class $\mathcal{H}$ is the pointwise minimal sample complexity attained by any of its learners, i.e., $m_{\mathrm{PAC},\mathcal{H}}(\epsilon, \delta) = \min_A m_{\mathrm{PAC},A}(\epsilon, \delta)$.*

As previously mentioned, transductive learning bears a close connection to PAC learning: see Asilis et al. [2024] and Dughmi et al. [2024] for further detail on their approximate equivalence.

**Lemma 3.19** (Asilis et al. [2024, Proposition 3.6]). *Let $\mathcal{X}$ be a domain, $\mathcal{Y}$ a label set, and $\mathcal{H} \subseteq \mathcal{Y}^{\mathcal{X}}$ a hypothesis class. Fix a loss function taking values in $[0,1]$. Then the following inequality holds for all $\epsilon, \delta \in (0,1)$ and the constant $e \approx 2.718$:*

$$m_{\mathrm{Trans},\mathcal{H}}(e \cdot (\epsilon + \delta)) \leq m_{\mathrm{PAC},\mathcal{H}}(\epsilon, \delta) \leq O\big(m_{\mathrm{Trans},\mathcal{H}}(\epsilon/2) \cdot \log(1/\delta)\big).$$

We now follow through on porting our results from the transductive model to the PAC model. The following is an immediate consequence of applying Lemma 3.19 to Theorems 3.9 and 3.7.

**Corollary 3.20.** *Let $\mathcal{X}$ be a domain, $\mathcal{Y}$ a label set, and $\mathcal{H} \subseteq \mathcal{Y}^{\mathcal{X}}$ a hypothesis class. Suppose that the loss function $\ell$ is bounded and satisfies either of the following conditions:*

- *$\ell$ is a metric on $\mathcal{Y}$, or*
- *$(\mathcal{Y}, d)$ is a compact metric space and $\ell$ is continuous with respect to this topology.*

*Then if all finite projections of $\mathcal{H}$ are learnable with realizable PAC sample function $m : (0,1)^2 \to \mathbb{N}$, $\mathcal{H}$ is learnable with sample complexity $O\big(m(\frac{\epsilon}{4e}, \frac{\epsilon}{4e}) \log(1/\delta)\big)$.*

Let us mention briefly that the connection between transductive learning and PAC learning may not be as tight in the agnostic case as in the realizable case. Through a straightforward use of Markov's inequality and a repetition argument, one can show that agnostic PAC sample complexities exceed transductive by at most a factor of $1/\epsilon$, but this is an unimpressive bound.

# 4 Conclusion

In this work, we studied the conditions under which the sample complexity of learning a class $\mathcal{H}$ can be detected by examining its finite projections. Notably, we established exact compactness results for transductive learning with a broad class of proper or continuous loss functions, across both realizable and agnostic learning. Using bounds relating the transductive and PAC models, we were able to transfer many of our results (in an approximate form) to realizable PAC learning. We leave as an open problem whether compactness of agnostic transductive sample complexities can fail by more than a factor of 2 for arbitrary (improper) metric losses. Additional future work includes better understanding the relationship between the transductive and PAC models in the agnostic case, and examining compactness for loss functions which do not satisfy any of our properness, metric, or continuity conditions (though they may be of somewhat limited interest in learning theory). It would also be of interest to study the compactness of error rates in settings other than supervised learning, such as online or unsupervised learning.

## Acknowledgments and Disclosure of Funding

Julian Asilis was supported by the Simons Foundation and by the National Science Foundation Graduate Research Fellowship Program under Grant No. DGE-1842487. Siddartha Devic was supported by the Department of Defense through the National Defense Science & Engineering Graduate (NDSEG) Fellowship Program. Shaddin Dughmi was supported by NSF Grant CCF-2009060. Vatsal Sharan was supported by NSF CAREER Award CCF-2239265 and an Amazon Research Award. Shang-Hua Teng was supported in part by the Simons Investigator Award from the Simons Foundation. Any opinions, findings, conclusions, or recommendations expressed in this material are those of the authors and do not necessarily reflect the views of any of the sponsors such as the NSF.

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

# A Proper metric spaces

Several of our results concern proper metric spaces. Let us expand briefly upon this condition, and present an equivalent definition.

**Definition A.1.** *A metric space $(\mathcal{Y}, d)$ is **proper** if either of the following equivalent conditions hold:*

1. *For all $Y \subseteq \mathcal{Y}$, if $Y$ is closed and bounded then it is compact.*

2. *For any $y \in \mathcal{Y}$ and $r > 0$, the closed ball $B_r(y) = \{y' \in \mathcal{Y} : d(y, y') \leq r\}$ is compact.*

Note that the conditions are indeed equivalent. That (1.) implies (2.) is immediate. Supposing (2.), note that any closed and bounded subset $Y$ is a closed subset of some closed ball, and thus compact.

We now discuss various sufficient conditions in order for a metric space to be proper.

**Lemma A.2.** *Let $(\mathcal{Y}, d)$ be a metric space. Any of the following conditions suffice to ensure that $(\mathcal{Y}, d)$ be a proper metric space.*

1. *$\mathcal{Y}$ is compact.*

2. *$\mathcal{Y}$ is finite.*

3. *$\mathcal{Y}$ is a closed subset of a proper metric space.*

*Proof.* If $\mathcal{Y}$ is compact, then its closed subsets are all compact. If $\mathcal{Y}$ is finite, then it is compact. If $\mathcal{Y}$ is a closed subset of a proper metric space $\mathcal{Y}'$, then its closed and bounded subsets are compact in $\mathcal{Y}'$ and thus compact in $\mathcal{Y}$. $\square$

Recall now that $\mathbb{R}^d$ endowed with the usual Euclidean norm is a proper metric space, owing to the Heine-Borel theorem. Invoking the equivalence of all norms on $\mathbb{R}^d$, it follows that $\mathbb{R}^d$ endowed with any norm enjoys the structure of a proper metric space.

**Corollary A.3.** *The following classes of metric spaces are proper:*

1. *All finite metric spaces.*

2. *All compact metric spaces.*

3. *$\mathbb{R}^d$, with any norm.*

4. *Any closed subset of $\mathbb{R}^d$, with any norm.*

Regarding necessary conditions for properness, note that all proper metric spaces are complete. Thus subsets of $\mathbb{R}^d$ which are not closed will not be proper, e.g., $\mathbb{Q} \subseteq \mathbb{R}^1$. See, e.g., Williamson and Janos [1987] for additional discussion and properties of proper metric spaces, which are sometimes referred to as Heine-Borel metric spaces.

# B Distribution-family Learning

The analysis of PAC learning with respect to more flexible distribution classes than the realizable and agnostic cases falls largely under the purview of *distribution-family learning* [Benedek and Itai, 1991]. Formally, a problem in distribution-family learning of a class $\mathcal{H} \subseteq \mathcal{Y}^{\mathcal{X}}$ is defined by a family of distributions $\mathbb{D}$ over $\mathcal{X}$, such that unlabeled datapoints are drawn from a distribution $D \in \mathbb{D}$ and labeled by a hypothesis $h \in \mathcal{H}$ (in the realizable case) or arbitrarily (in the agnostic case).

Notably, distribution-family learning has infamously resisted any characterization of learnability, combinatorial or otherwise, for the 40 years since its inception. In fact, there is some evidence to suggest that no such characterization may exist [Lechner and Ben-David, 2023]. Furthermore, it is a setting in which uniform convergence fails to characterize learning, rendering ineffective many of the standard and most celebrated techniques of learning theory.

Nevertheless, we now demonstrate that compactness sheds light on the problem of distribution-family learning, at least for the case of *well-behaved* distribution classes.

**Definition B.1.** *A family of distributions $\mathbb{D}$ over a set $\mathcal{Z}$ is **well-behaved** if whenever $S = (z_1, \ldots, z_n)$ lies in the support of some $D \in \mathbb{D}$, then $\mathrm{Unif}(S)$, the uniform distribution over $S$, lies in $\mathbb{D}$ as well.*

Definition B.1 is sufficiently flexible that we may apply it to distribution-family learning with $\mathcal{Z} = \mathcal{X}$ or to PAC learning over arbitrary distribution classes $\mathbb{D}$ with $\mathcal{Z} = \mathcal{X} \times \mathcal{Y}$. Though it may appear overly restrictive at first glance, note that well-behavedness is satisfied not only for ordinary PAC learning in the agnostic and realizable cases, but also for learning of partial concept classes in the realizable case [Alon et al., 2022] and for the EMX learning of Ben-David et al. [2019]. In particular, though EMX learning is not presented as a supervised learning problem in Ben-David et al. [2019], it can be seen as a binary classification problem over a domain $\mathcal{X}$ for which $\mathcal{H}$ consists of those functions outputting finitely many 1's and $\mathbb{D}$ contains all realizable, discrete distributions placing all $\mathcal{Y}$-mass on the label 1.

Crucially, well-behavedness permits us to study PAC learning by way of transductive error.

**Proposition B.2.** *Let $\mathcal{H} \subseteq \mathcal{Y}^{\mathcal{X}}$ be a hypothesis class and $\mathbb{D}$ a well-behaved family of distributions which are realizable (i.e., $\inf_{\mathcal{H}} L_D(h) = 0 \; \forall D \in \mathbb{D}$). Fix a loss function taking values in $[0, 1]$. Then the following inequality holds for all $\epsilon, \delta \in (0, 1)$ and the constant $e \approx 2.718$:*

$$m_{\mathrm{Trans}, \mathcal{H}}(e \cdot (\epsilon + \delta)) \leq m_{\mathrm{PAC}, \mathcal{H}}(\epsilon, \delta) \leq O\big(m_{\mathrm{Trans}, \mathcal{H}}(\epsilon/2) \cdot \log(1/\delta)\big).$$

*Proof.* The proof is nearly identical to that of [Asilis et al., 2024, Proposition 3.6]. In particular, let $m_{\mathrm{Exp}, \mathcal{H}}$ denote the sample complexity of learning $\mathcal{H}$ in the expected error regime (i.e., $m_{\mathrm{Exp}, \mathcal{H}}(\epsilon)$ equals the number of datapoints needed to incur expected error at most $\epsilon$). Then we have

$$m_{\mathrm{Exp}, \mathcal{H}}(\epsilon + \delta) \leq m_{\mathrm{PAC}, \mathcal{H}}(\epsilon, \delta) \leq O\big(m_{\mathrm{Exp}, \mathcal{H}}(\epsilon/2) \cdot \log(1/\delta)\big).$$

The first inequality follows immediately from the fact that the loss function is bounded above by 1. The second inequality follows from a repetition argument, i.e., a learner attaining expected error $\leq \epsilon/2$ on samples of size $n$ can be boosted to attain expected error $\leq \epsilon$ with probability $\geq 1 - \delta$ by using an additional factor of $O(\log(1/\delta))$ many samples, as described in [Danely and Shalev-Shwartz, 2014].

We now show that $m_{\mathrm{Exp}, \mathcal{H}}(\epsilon)$ and $m_{\mathrm{Trans}, \mathcal{H}}(\epsilon)$ are essentially equivalent, i.e., that

$$m_{\mathrm{Exp}, \mathcal{H}}(\epsilon) \leq m_{\mathrm{Trans}, \mathcal{H}}(\epsilon) \leq m_{\mathrm{Exp}, \mathcal{H}}(\epsilon/e).$$

The first inequality follows inequality from a standard leave-one-out argument of Haussler et al. [1994]. For the second inequality, let $A$ be an optimal learner in the expected error paradigm, and note that a transductive learning instance $S = (x_i, y_i)_{i \in [n]}$ can be solved as follows: Given training set $S_{-j}$ and test point $x_j$, return $A(T)(x_j)$, where $T$ is a sample of points drawn independently and uniformly at random from $S_{-j}$. The crucial observation is that $T$ mimics an i.i.d. sample drawn from $S$, as such a sample will happen to avoid $x_i$ with probability $\geq 1/e$. Thus the transductive error of our described learner can exceed that of $A$ by at most a factor of $e$. $\qquad \square$

Note that Proposition B.2 holds for the natural definition of transductive learning with respect to $\mathbb{D}$, i.e., in which the adversary must select a sequence of unlabeled datapoints $S \in \mathcal{X}^n$ which lie in the support of some $D \in \mathbb{D}$. It is now immediate from the proof of Theorem 3.6 that distribution-family learning is a setting in which the transductive sample complexity of learning is $\mathcal{H}$ equals the sample complexity of learning its most challenging finite projections. In the following, we let $\mathbb{D}|_X \subseteq \mathbb{D}$ denote the distributions of $\mathbb{D}$ which place full measure on $X \subseteq \mathcal{X}$.

**Theorem B.3.** *Let $\mathcal{X}$ be an arbitrary domain, $\mathcal{Y}$ a proper metric space, and $\mathcal{H} \subseteq \mathcal{Y}^{\mathcal{X}}$ a hypothesis class. Let $\mathbb{D}$ be a family of well-behaved, realizable distributions. Then the following are equivalent for any $m : \mathbb{R}_{>0} \to \mathbb{N}$:*

1. *$\mathcal{H}$ is learnable with respect to $\mathbb{D}$ with transductive sample function $m$.*

2. *For any finite $X \subseteq \mathcal{X}$ and finite $\mathcal{H}' \subseteq \mathcal{H}|_X$, $\mathcal{H}'$ is learnable with respect to $\mathbb{D}|_X$ with transductive sample function $m$.*

As in Section 3.4, Proposition B.2 applied to Theorem B.3 immediately yields an almost-exact form of compactness for distribution-family PAC learning with realizable and well-behaved distribution classes. This demonstrates that the learnability of even problems as exotic as EMX learning can be detected by examining all their finite projections, provided that no restrictions are placed upon

learners. In Ben-David et al. [2019], however, learners were required to only emit hypotheses in $\mathcal{H}$. Our work demonstrates that the nature of their undecidability result — in which the learnability of a class $\mathcal{H}$ is determined entirely by its cardinality, despite all its projections being easily learned — could otherwise not appear in supervised learning with metric losses.

## C   Omitted proofs

### C.1   Proof of Theorem 3.8

*Proof.* Set $\mathcal{X} = \mathbb{N}$ and let $\mathcal{Y}$ be the metric space defined as follows. $\mathcal{Y} = R \cup S$, where $R$ is an infinite set whose points are all distance 2 apart. $S$ is an infinite set whose points are indexed by finite subsets of $R$, e.g., as in $s_{R'} \in S$ for finite $R' \subseteq R$. For all such $R'$, define $s_{R'}$ to be distance 1 from the elements of $R'$, distance 2 from the other elements of $R$, and distance 1 from all other points in $S$. Note that $\mathcal{Y}$ indeed forms a metric space as its distance function is positive-definite, symmetric, and only uses the non-zero values of 1 and 2. (In particular, the triangle inequality is satisfied as an automatic consequence of the latter fact.) Now fix an $\bar{r} \in R$ and $k \in \mathbb{N}$. Define $\mathcal{H} \subseteq \mathcal{Y}^{\mathcal{X}}$ to consist of all those functions which output $\bar{r}$ on all inputs $x > k$. Notably, any $h \in \mathcal{H}$ may take arbitrary values on $x \leq k$.

Let us first analyze the sample complexity of learning a finite projection of $\mathcal{H}$. Fix finite $X \subseteq \mathcal{X}$ and finite $\mathcal{H}' \subseteq \mathcal{H}|_X$. Then, as $X$ and $\mathcal{H}'$ are each finite, the images of all the $\mathcal{H}'$ are contained in a finite set $Y' \subseteq \mathcal{Y}$. Let $Y'$ decompose as $Y' = R' \cup S'$ with $R' \subseteq R, S' \subseteq S$. Then the following learner attains error $\leq \frac{k}{n}$ on instances of size $n$:

$$A : (\mathcal{X} \times \mathcal{Y})^{<\omega} \longrightarrow \mathcal{Y}^{\mathcal{X}}$$

$$S \longmapsto x \mapsto \begin{cases} s_{R'} & x \leq k, \\ \bar{r} & x > k. \end{cases}$$

In particular, for any $x \leq k$ and $h \in \mathcal{H}'$, $d(A(S)(x), h(x)) \leq 1$, while for any $x > k$, $A(S)$ emits the correct prediction by definition of $\mathcal{H}$. Furthermore, we may assume without loss of generality that transductive learning instances do not contain repeated datapoints, as these only lessen the difficulty of learning. Thus any sample $S$ has at most $k$ unlabeled datapoints in $[k]$, and the previous analysis demonstrates that $\mathcal{H}'$ can be learned with error $\leq \frac{k}{n}$ when $|S| = n$.

On the other hand, for the case of learning $\mathcal{H}$ itself, there exist $n$ such that the worst-case error incurred by any learner on $|S| = n$ is at least $\frac{2k}{n}$. In particular, take $n = 1$. As $\mathcal{Y}$ has radius 2 — and furthermore for any $y \in \mathcal{Y}$ there exists $y' \in \mathcal{Y}$ with $d(y, y') = 2$ — transductively learning $\mathcal{H}$ with $n = 1$ is guaranteed to incur an error of at least 2 in the worst case. (That is, by taking $S = \{x\}$ with $x \leq k$.) Thus, for $n = 1$ and any finite projection $\mathcal{H}'$ we have $\xi_{\mathcal{H}'}(n) \leq 1$ and $\xi_{\mathcal{H}}(n) \geq 2$, and the claim follows. $\qquad\square$

### C.2   Proof of Theorem 3.9

Before commencing with the proof, let us establish some terminology and a supporting lemma. Fix a hypothesis class $\mathcal{H}$ along with $S \in \mathcal{X}^n$ and $\epsilon > 0$. Recall the collection of functions $R = \mathcal{H}|_S$ and variables $L = \bigcup_{S' \subseteq S, |S'| = n-1} \mathcal{H}|_{S'}$ which capture the structure of transductive learning on instances of the form $\{(S, h)\}_{h \in \mathcal{H}}$, as described in the proof of Theorem 3.6. In this setting, for any number $\alpha > 0$, we will let $\alpha$-*apportionment* refer to a vector $\nu = (x_1, \ldots, x_n)$ with $x_i \geq 0$ and $\sum x_i = \alpha$. We may suppress $\alpha$ and refer simply to apportionments.

Given an $\alpha$-apportionment $\nu$ for a node $r \in R$, we will say that an assignment of variables in $L$ *satisfies* this apportionment if the output of $r$ as a function decomposes according to $\nu$. More explicitly, recall that $r = (y_1, \ldots, y_n)$ depends upon the variables $\{\ell_i\}_{i \in [n]} \subseteq L$, where $\ell_i = (y_1, \ldots, y_{i-1}, ?, y_{i+1}, \ldots, y_n)$ and

$$r(\ell_1, \ldots, \ell_n) = \frac{1}{n} \sum_{i \in [n]} d(y_i, \ell_i).$$

Then an assignment of variables $\{\ell_i\}_{i \in [n]} \to \mathcal{Y}$ satisfies $\nu = (x_1, \ldots, x_n)$ if $\frac{1}{n} d(y_i, \ell_i) \leq x_i$ for all $i \in [n]$. Intuitively, $\nu$ tracks the manner in which $r$ produces its output. Similarly, given

apportionments for various nodes in $\mathbb{R}$, we say that a given assignment of variables *satisfies* the apportionments if it satisfies each of them at once.

With a slight abuse of terminology, we say that a node $r \in R$ without an apportionment is *satisfied* by an assignment of variables $L \to \mathcal{Y}$ if its output is at most $\epsilon$ under the assignment. There should be no risk of confusion, as it will always be clear whether a given node $r \in R$ is endowed with an apportionment or not. In a similar fashion to Theorem 3.6, we say a *partial assignment of apportionments* is an assignment of apportionments to a subset of $R$. An assignment which happens to be total is referred to as a total assignment of apportionments. A partial assignment $\phi$ is *satisfiable* with respect to $R' \subseteq R$ if there exists an assignment of variables $L \to \mathcal{Y}$ such that all nodes in $R'$ are satisfied. (I.e., have their apportionments satisfied if equipped with one, and are otherwise simply maintained below $\epsilon$.) We say $\phi$ is *finitely satisfiable* if it is satisfiable with respect to all finite subsets of $R$.[3]

**Lemma C.1.** *Let $\mathcal{X}$ be a domain, $\mathcal{Y}$ a metric space, and $\mathcal{H} \subseteq \mathcal{Y}$ a hypothesis class. Fix $S \in \mathcal{X}^n$ and the corresponding collections of functions $R = \mathcal{H}|_S$ and variables $L = \bigcup_{S' \subseteq S, |S'|=n-1} \mathcal{H}|_{S'}$. Let $\delta > 0$. If $R$ is finitely satisfiable, when none of its elements are endowed with apportionments, then there exists a total assignment of $(\epsilon + \delta)$-apportionments to $R$ which is finitely satisfiable.*

*Proof.* We appeal to Zorn's lemma. The argument relies crucially upon the fact that any partial assignment of $(\epsilon + \delta)$-apportionments which is finitely satisfiable can be augmented by assigning an additional apportionment to an unassigned function in $R$.

> **Lemma C.2.** *Let $\phi$ be a partial assignment of $(\epsilon+\delta)$-apportionments to $R$ which is finitely satisfiable and leaves a function in $R$ unassigned. Then one such unassigned variable can receive an $(\epsilon + \delta)$-apportionment while preserving finite satisfiability.*
>
> *Proof.* Fix an unassigned variable $r \in R$, along with a finite collection of nodes $R' \subseteq R$. Let $R' = R_0 \cup R_1$, where the nodes in $R_0$ are unassigned by $\phi$ (i.e., need only be maintained below $\epsilon$), and those in $R_1$ are assigned by $\phi$ (i.e., need be maintained below $\epsilon + \delta$ and furthermore have their apportionments respected). Let $A(R')$ denote the collection of variable assignments $L \to \mathcal{Y}$ which satisfy all nodes in $R'$ along with $r$. By the supposition that $\phi$ is finitely satisfiable, $A(R')$ is non-empty.
>
> Now let $\Phi(R')$ denote all $\epsilon$-apportionments for $r$ that are satisfied by an assignment in $A(R')$. Informally, these are the $\epsilon$-apportionments with which we could endow $r$, if we only needed to consider the nodes $R'$. Then let $\mathrm{cl}(\Phi(R'))$ denote the closure of $\Phi(R')$ in $\mathbb{R}^n$, and consider the family of sets
>
> $$\mathcal{I} = \{\mathrm{cl}(\Phi(R')) : R' \subseteq R, |R'| < \infty\}.$$
>
> Each set $I \in \mathcal{I}$ is compact, as it is closed and bounded. Furthermore, finite intersections of such sets are non-empty, as
>
> $$\bigcap_{i=1}^{k} \mathrm{cl}(\Phi(R_i)) \supseteq \bigcap_{i=1}^{k} \Phi(R_i) \supseteq \Phi\left(\bigcup_{i=1}^{k} R_i\right) \neq \emptyset.$$
>
> Then there exists an $\epsilon$-apportionment $\nu \in \bigcap \mathrm{cl}(\Phi(R'))$, where the intersection ranges over all finite subsets of $R$. As we took the closures of the sets $\Phi(R')$, this does not suffice to guarantee us that $\bigcap \Phi(R')$ is non-empty. Let us now increase each of the entries of $\nu$ by an arbitrarily small amount, say $\delta/n$, and call the resulting $(\epsilon + \delta)$-apportionment $\nu^*$.
>
> For any finite $R' \subseteq R$, recall that $\nu \in \mathrm{cl}(\Phi(R'))$, meaning there exist assignments satisfying $R'$ and $r$ which induce $\epsilon$-apportionments on $r$ of arbitrarily small proximity to $\nu$. As $\nu^*$ strictly exceeds $\nu$ in each coordinate, then there exists an assignment satisfying $R'$ which also satisfies $\nu^*$. And $R'$ was chosen arbitrarily, so the claim follows. $\square$

---

[3]Notably, this is a pointwise condition, not a uniform one. The satisfying assignments are permitted to vary across the finite subsets of $R$.

Consider now the poset $\mathcal{P}$ whose elements are finitely satisfiable partial assignments of $(\epsilon + \delta)$-apportionments to $R$. The partial ordering is such that $\phi_1 \leq \phi_2$ if $\phi_2$ agrees with all assignments of apportionments made by $\phi_1$ and perhaps assigns additional apportionments. It is straightforward to see that chains in $\mathcal{P}$ have upper bounds: let $\mathcal{C} \subseteq \mathcal{P}$ be a chain and define $\phi_{\mathcal{C}}$ to be the "union" of assignments in $\mathcal{C}$, i.e., $\phi_{\mathcal{C}}$ leaves $r \in R$ unassigned if all $\phi \in \mathcal{C}$ leave $r$ unassigned, otherwise it assigns $r$ to the unique apportionment used by the assignments in $\mathcal{C}$.

Certainly $\phi_{\mathcal{C}}$ serves as an upper bound for $\mathcal{C}$, provided that $\phi_{\mathcal{C}} \in \mathcal{P}$. To see that $\phi_{\mathcal{C}} \in \mathcal{P}$, fix a finite set $R' \subseteq R$. Suppose $S \subseteq R'$ is the collection of nodes in $R'$ which receive $(\epsilon + \delta)$-apportionments from $\phi_{\mathcal{C}}$. Each $s \in S$ receives an apportionment from a $\phi_s \in \mathcal{C}$. Then, as $\mathcal{C}$ is a chain and $\{\phi_s\}_{s \in S} \subseteq \mathcal{C}$ is a finite set, there exists an $s' \in S$ with $\phi_{s'} = \max\{\phi_s\}_{s \in S}$. By the definition of the partial order with which we endowed $\mathcal{P}$, $\phi_{s'}$ then agrees exactly with $\phi_{\mathcal{C}}$ when restricted to the set $R'$. As $\phi_{s'} \in \mathcal{P}$, $R'$ is satisfiable with respect to $\phi_{s'}$ and thus satisfiable with respect to $\phi_{\mathcal{C}}$. As $R$ was selected arbitrarily, we have that $\phi_{\mathcal{C}}$ is finitely satisfiable, meaning $\phi_{\mathcal{C}} \in \mathcal{P}$ and chains indeed have upper bounds.

Then, invoking Zorn's lemma, $\mathcal{P}$ contains a maximal element $\phi_{\max}$. By Lemma C.2, it must be that $\phi_{\max}$ does not leave any function in $R$ unassigned, otherwise it could be augmented with an additional apportionment, contradicting maximality. Thus there exists a *total* assignment of $(\epsilon + \delta)$-apportionments which is finitely satisfiable, completing the argument. $\qquad\square$

We are now equipped to prove Theorem 3.9 itself.

*Proof.* Let $\mathcal{H}$ be as in the theorem statement, fix an $n \in \mathbb{N}$, and let $\epsilon = \xi(n)$. We will exhibit a learner $\mathcal{A}$ for $\mathcal{H}$ attaining error at most $2\epsilon + \delta$ on samples of size $n$, for arbitrarily small $\delta > 0$. To this end, fix one such $\delta > 0$ and an $S \in \mathcal{X}^n$, and recall the system of functions $R$ and variables $L$ which capture learning on transductive instances of the form $\{(S, h)\}_{h \in \mathcal{H}}$. That is, we represent the functions in $R$ as $R = \mathcal{H}|_S$ and the variables in $L$ as $L = \cup_{S' \subseteq S, |S'|=n-1} \mathcal{H}|_{S'}$. As described in the proof of Theorem 3.6, we may suppress the unlabeled datapoints in the definitions of $R$ and $S$, and represent each $r \in R$ as an element of $\mathcal{Y}^n$ and each $\ell \in L$ as an element of $(\mathcal{Y} \cup \{?\})^n$ with exactly one "?". Recall too that each function $r \in R$, represented by $(y_1, \ldots, y_n) \in \mathcal{Y}^n$, depends upon the variables $\ell_1, \ldots, \ell_n \in L$, where

$$\ell_i = (y_1, \ldots, y_{i-1}, ?, y_{i+1}, \ldots, y_n).$$

Upon assigning each such variable $\ell_i$ to an element of $\mathcal{Y}$ (semantically, a completion of its "?" entry corresponding to a query at test time) the function $r$ outputs the value

$$\frac{1}{n} \cdot \sum_{i=1}^{n} d_{\mathcal{Y}}(y_i, \ell_i).$$

The central observation is that defining a learner $\mathcal{A}$ for $\mathcal{H}$ which attains error $\leq 2\epsilon + \delta$ amounts precisely to assigning each variable in $L$ to a value in $\mathcal{Y}$ such that the functions in $R$ are all maintained below $2\epsilon + \delta$. By the premise of the theorem, this collection of functions and variables is finitely satisfiable. That is, for each finite $R' \subseteq R$, there exists an assignment of all variables $L \to \mathcal{Y}$ such that all functions in $R'$ are maintained below $\epsilon$.[4] Then, by Lemma C.1, there exists an assignment of $(\epsilon + \delta/3)$-apportionments to each function in $R$ which is finitely satisfiable.

Now fix a node $\ell \in L$: we will demonstrate how to assign it to a value of $\mathcal{Y}$. Note that $\ell$ influences a potentially infinite collection of functions $\{r_i\}_{i \in I} \subseteq R$. Each such function has an apportionment of error for $\ell$. Call these values $\{\lambda_i\}_{i \in I}$. Recall that each node $r_i$ for $i \in I$ computes its error incurred on $\ell$ relative to a label $y_i \in \mathcal{Y}$. Now choose an index $N \in I$ such that $\lambda_N - \inf_I \lambda_i \leq \delta/3$. We then set $\ell = y_N$.

In order to analyze this assignment, fix an arbitrary $i \in I$ and consider the set $R' = \{r_i, r_N\}$. By finite satisfiability of our system of $(\epsilon + \delta/3)$-apportionments, there exists an assignment of variables

---

[4]Strictly speaking, the definition of error rate involves an infimum, meaning we are only guaranteed that the functions in $R'$ can be maintained arbitrarily close to $\epsilon$. It is straightforward to see that this suffices for our purposes, however, as Lemma C.1 results in the addition of an arbitrarily small term to $\epsilon$ anyway.

$L \to \mathcal{Y}$ satisfying the functions in $R'$. Let $y^*$ be the value of variable $\ell$ in this assignment. We have:

$$
\begin{aligned}
d_{\mathcal{Y}}(\ell, y_i) &= d_{\mathcal{Y}}(y_N, y_i) \\
&\leq d_{\mathcal{Y}}(y_N, y^*) + d(y^*, y_i) \\
&\leq \lambda_N + \lambda_i \\
&\leq \left( \inf_I \lambda_i + \frac{\delta}{3} \right) + \lambda_i \\
&\leq 2 \cdot \lambda_i + \frac{\delta}{3}.
\end{aligned}
$$

Now assign all variables in $L$ in this manner, and consider an arbitrary node $r \in R$ with error apportionment $\lambda_1, \ldots, \lambda_n$ for each of the variables upon which it depends. Then, using the above analysis, $r$ evaluates to at most

$$
\frac{1}{n} \sum_{i=1}^{n} \left( 2 \cdot \lambda_i + \frac{\delta}{3} \right) \leq 2 \cdot \left( \epsilon + \frac{\delta}{3} \right) + \frac{\delta}{3} = 2 \cdot \epsilon + \delta,
$$

completing the argument. $\qquad\square$

## C.3  Proof of Lemma 3.12

*Proof.* Fix a bipartite graph $G = (L \cup R, E)$ such that all nodes in $R$ have finite degree and $G$ is finitely $R$-matchable. We will demonstrate the existence of a subgraph $G' = (L \cup R, E')$ of $G$ such that $G'$ is finitely $R$-matchable.

First note that $G$ is finitely $R$-matchable if and only if P. Hall's condition holds for all finite subsets $R'$ of $R$. That is, if and only if $|N(R')| \geq |R'|$, where $N(R')$ denotes the set of neighbors $R'$ has in $L$, by Hall [1935]. We refer to a finite subset $R'$ of $R$ as a *blocking set* if $|N(R')| = |R'|$. We say a node $\ell \in L$ is contained in a blocking set if there exists a blocking set $R'$ such that $\ell \in N(R')$.

Now consider all nodes in $L$; if there exists a node $\ell \in L$ such that $\ell$ is not in any blocking set, then it can be removed from $L$ while preserving Hall's condition in the graph (i.e., while preserving finite $R$-matchability). Repeatedly removing nodes from $L$ in this way and applying Zorn's lemma, we arrive at a collection of nodes $L' \subseteq L$ such that Hall's condition is preserved and each node in $L$ is contained in a blocking set. Call the resulting graph $G'$.

Now fix a node $\ell \in L'$ and pick a blocking set $T \subseteq R$ containing $\ell \in L'$. We remove all edges incident to $\ell$ which are not incident to $R'$. Let us demonstrate that the remaining graph $G''$ remains finitely $R$-matchable, i.e., satisfies Hall's condition. Suppose not, so that there exists a finite set $S \subseteq R$ violating Hall's condition. Then, as $G'$ satisfies Hall's condition, it must be that $\ell$ is incident to $S$ in $G'$ but not in $G''$. Furthermore, it must be that $S$ is a blocking set in $G'$, as was $R$. Then consider the set $S \cup T$ in $G'$. We have:

$$
\begin{aligned}
|N(S \cup T)| &= |N(S) \cup N(T)| \\
&\leq |N(S)| + |N(T)| - |N(T) \cap N(S)| \\
&\leq |N(S)| + |N(T)| - |N(T \cap S)| - |\{\ell\}| \\
&\leq |S| + |T| - |T \cap S| - 1 \\
&< |S \cup T|
\end{aligned}
$$

producing contradiction with Hall's condition for $G'$. Note that the third line makes use of the fact that $|N(T) \cap N(S)| \supseteq |N(T \cap S)| \sqcup \{\ell\}$, as clearly $|N(T) \cap N(S)| \supseteq |N(T \cap S)|$ and furthermore $\ell \in \big( N(T) \cap N(S) \big) \setminus N(T \cap S)$ owing to the fact that $\ell$ is incident to both $T$ and $S$ but not due to a node in $T \cap S$ (otherwise $\ell$ would have remained incident to $S$ in $G''$).

We are thus permitted to perform the operation on any single node of $L'$ to make its degree finite while preserving Hall's condition. As any failure of Hall's condition can be detected by way of finitely many nodes in $L'$, it follows that we can do so for all nodes of $L'$ in concert. The resulting graph is a subgraph of $G$ which is finitely $R$-matchable and for which all nodes have finite degree, as desired. $\qquad\square$

