# OpenReview forum: "Transductive Learning is Compact"
_NeurIPS.cc/2024/Conference — NeurIPS 2024 poster_

### Official Review · Reviewer_JpD2 · 2024-06-25

**Soundness:** 3
**Presentation:** 3
**Contribution:** 3
**Rating:** 7
**Confidence:** 2

**Summary:**

The paper looks at the question of whether or not the transductive sample complexity is compact in the sense that the transductive sample complexity globally can be deduced from understanding the tranductive sample complexity of finite instances of transductive learning. More formally transdictive learning is the setting where the learner is revealed a labelled dataset $S_i$ with one variables $x_i$ missing and is then querried to predict the label of the  $x_i$, where all the labels $y_1,\ldots,y_n$ are determined by some $h\in\mathcal{H}$ i.e. $y_{j}=h(x_j)$. The goal is now create a learned $A$ which takes as input the $S_i$ and $x_i$ and for some loss $L: \mathcal{Y}\times \mathcal{Y}\rightarrow R_{0\geq }$,  minimize $L(A,S,h)=\sum_{i}^{n} L(A(S_i,x_i),h(x_i))$ for any $h\in \mathcal{H}$ and $S\subset \cup_{i=1}^{\infty}\mathcal{X}^{i}$. This can also be seen as a game where a adversary picks $h\in\mathcal{H}$ and $S\subset  \cup_{i=1}^{\infty} \mathcal{X}^{i} $ and then look at the expected value of the loss of a learner which is given $S_i$ and propmt the label of $x_i$ for a random $i\sim |S|$. The sample complexity of transductive learning is now for $\varepsilon$ defined as

 $$m(\varepsilon)=\min\{m\in\mathbf{N}: \inf_{A} \sup_{h\in\mathcal{H},S\in\mathcal{X}^{m}} L(A,S,h)\leq \varepsilon\}$$



The paper shows that the transductive sample complexity is compact in the following sense: For suitable loss functions(metric losses, contious losses on compact spaces, the zero one loss) the transductive sample complexity of a hypothesis class $\mathcal{H}$ $m()$ is the same as for any finite $X\subset \mathcal{X}$ and finite $H'\subset \mathcal{H}|_{X}$ having transductive sample complexity $m()$. They also show this for the agnostic setting of transductive learning, and distribution-families and furthermore show how PAC-learning can be studied from a point of view of finite projections of $\mathcal{H}$. The authors also give an example of where the transductive sample complexity is not compact - but close to.

**Strengths:**

Originality:

The paper states that they are the first to discover exact connection of the compactness of transductive sample complexity in the realizable and agnostic setting.


Quality and Clarity:

The paper is well written and things are nicely defined and explained.

Significance:

The idea about connecting tranductive sample complexity seems novel and interesting.

**Weaknesses:**

.

**Questions:**

Line 48: Why is $\mathcal{H}_{S}$ nessarily finite? What if $\mathcal{Y}$ is infinite?
\newline

Line 174-175: In the definition of completable partial sisignment: Is it that there exist an assignment of the unassigned variables such that for all $r\in R'$ this assignment is such that $r$ is less than $\varepsilon$ or is it for all $r\in R'$ there exists and assigment of the unassigned variables such that $r$ is less than $\varepsilon$?

Line 195: Why isn't it $|R'|<\infty$ instead of $|R|\leq \infty$ so excluding $=$?

Line 207: Do you have to be in $\mathcal{P}$ to be an upper bound?

Line 212-213: Why isn't: That is, there exists $\phi_{j} \in C$ which agrees with $\phi_{C}$ in
its action on $l_{1}, \ldots, l_{i}$. As $\phi_{j}\in P$, it must be that $\phi_{j}$ is completable with respect to $R'$ .i.e. there exists assignments to $l_{i+1}\ldots,l_{m}$ "such that $\leq \varepsilon$", but since $l_{i+1},\ldots,l_{m}$ are free variables of $\phi_{C}$ and $\phi_{C}$ upper bound $\phi_{j}$ so especially agree $\phi_{j}$ on $l_{1},\ldots,l_{i}$ the variables $l_{i+1},\ldots,l_{m}$ can also be assigned such that $\phi_{C}$ is
also completable with respect to $R'$ with the same assignment of the variables $l_{i+1},\ldots,l_{m}$ as $\phi_{j}$. Sorry if it is saying the same.

Line 216: total in what sense?

Theorem 3.6: Can you please provide the set of variables $L$ and $R$ in the proof of Theorem 3.6?(Formally)

Line 244: Why is the $\inf$ always attained?

Line 246: Why does $r$ reflects bounded sets- because it is a function of a norm on $\mathcal{Y}$ - so cant be to fare $(O(\varepsilon))$ from the fix argument - i.e. in a bound ball around that?

Line 331: The condition about disjointness in a R-matching I struggle with - could you try to explain why each $R$ only has one $l$ insident, when it has a degree larger than $1$?

Line 352-354: Out of curiosity, why is it that 2 and 3 are not combined?

(not a very concrete question and out of curiosity so if time is limited not need to be answered): The finding is also very broth and theoretical (a strength). I am curious if the idea of studying the transductive sample complexity "locally" instead of "globally" have been used before in the analysis of the transductive sample complexity - the authors meantions in the related work section the $\gamma-OIG$ - is the analysis of $\gamma-OIG$ for instance done in this way.

**Limitations:**

Yes.

---

> ### Author Rebuttal · Authors · 2024-08-05
>
> Thank you for your time and attention in reviewing the paper. We will address your questions in order.
>
> Line 48: $H|_S$ can indeed be infinite when $Y$ is infinite. For this reason, we refer only to *finite subsets* of $H|_S$ as finite projections of $H$. That is, a finite projection of $H$ refers to both restricting to a finite collection of points in the domain *and* restricting to finitely many behaviors on those points (lines 45-48). We will add further clarification to the beginning of Section 1.1.
>
> Line 174-175: It is defined as the first option you mentioned: there exists an assignment of unassigned variables satisfying all functions in $R’$. I.e., the assignment is permitted to depend upon the choice of $R’$, but it may *not* depend upon particular functions in $R’$. We will clarify this further in the next draft.
>
> Line 195: That is a typo, the inequality should be strict – thank you!
>
> Line 207: Yes, an element must be in the poset $P$ in order to be an upper bound, as it must be in $P$ in order to be comparable to any elements in $P$.
>
> Line 212-213: Yes, you are exactly correct! Unfortunately we must be slightly terse at times owing to the page limit. We will further clarify this step in the next version of the paper.
>
> Line 216: Total in the sense of assigning *all* variables, i.e., not leaving any variables unassigned. We will clarify this in the paper.
>
> Theorem 3.6: Formally, $L$ is a set of variables which are valued in $Y$ and which are indexed/parameterized in the following manner: `L = \cup_{S' \subseteq S, |S'| = n-1} H|_{S'}`. So each variable $\ell \in L$ is represented by a tuple of the form $(y_1, \ldots, y_{i-1}, ?, y_{i+1}, \ldots, y_n)$.
>
> $R$ is a collection of functions, indexed/parameterized by $R = H|_S$. The function $r \in R$ which is represented by $(y_1, \ldots, y_n)$ depends upon the $n$ variables $\ell \in L$ whose representing vectors agree with that of $r$ except for a “?” in one of its entries. (Formally, the representing vectors of $r$ and $\ell$ have the same values in the non-"?" entries of $\ell$.) Lastly, we define the output of $r$ on these variables $\ell_1, \ldots, \ell_n$ as $\frac 1n \sum d(y_i, \ell_i)$.
>
> Line 244: By assuming condition (2.), we have that all finite projections of H can be learned to error $\leq \epsilon$ using $n = m(\epsilon)$ many samples. This is equivalent to condition (2.) of Theorem 3.3 for the system of variables and functions we describe, allowing us to conclude condition (1.). In that sense, we do not need to worry about infimums.
>
> Line 246: Exactly: the pre-image of bounded sets must be bounded because the function uses the norm on Y. (E.g., the pre-image of $[0, \epsilon]$ contains only points within distance $\epsilon$ of $r = (y_1, \ldots, y_n)$ in each coordinate.)
>
> Line 331: Each node in $R$ is permitted to have multiple incident nodes in $L$. An $R$-matching, however, consists of only one *choice* of incident edge for each node in $R$. The requirement is that these chosen edges be disjoint, i.e., that they each connect to distinct nodes in $L$. The essence of Theorem 3.13 is that the existence of such $R$-matchings can be detected locally. That is, $R$-matchings exist precisely when $R’$-matchings exist for all finite $R’ \subseteq R$.
>
> Line 352-354: At the level of information, it would indeed be equivalent to combine steps 2 and 3 (or even to remove step 2 entirely). We mention step 2 primarily to emphasize that the learner has access to all unlabeled datapoints in $S$, call them $S_X$. This allows it to, for instance, construct the OIG of $H$ on $S_X$ after step 2 and orient it optimally, and then use the information of step 3 to get a prediction for the unlabeled test point. This is often how transductive learning is described in the literature, but in the absence of step 2 the learner could equivalently just wait until after step 4 to see all unlabeled data, construct the OIG, orient it, and perform inference all in one go. It is primarily a matter of preference/exposition.
>
> Local transductive sample complexity: Excellent question. One-inclusion graphs and the transductive model are in some sense intrinsically local, as they only study the performance of learners on finite regions of the domain. (As opposed to the PAC model, which considers performance with respect to distributions with possibly infinite support.) Our results can be viewed as stating that the transductive model can be made even *more* local by restricting focus to finite regions of the domain and only finitely many behaviors on the domain. (I.e., to finite subgraphs of one-inclusion graphs.) To our knowledge, this is as local a perspective as one could hope to achieve.
>
> The $\gamma$-OIG is defined and analyzed using largely the same perspective as for classical OIGs. There, the key difference is that the loss function is not the 0-1 loss but rather the absolute loss on the interval $Y = [0, 1]$. The key idea behind the $\gamma$-OIG is to turn this continuous loss into a discrete one by thresholding: losses below $\gamma$ are rounded to 0 and losses above $\gamma$ are rounded to 1. ($\gamma$ is thought of as a scale parameter ranging from 1 to 0.) This permits the authors to simplify the problem and to use ideas and techniques from discrete math (e.g., graph theory).
>
> Thank you again for your detailed questions and comments! They will all be taken into account to improve the paper.

---

### Official Review · Reviewer_B4fj · 2024-07-11

**Soundness:** 3
**Presentation:** 3
**Contribution:** 3
**Rating:** 6
**Confidence:** 3

**Summary:**

The paper studies transductive learning with general real-valued loss functions. The paper shows that:
(1) for proper metric loss functions and continuous loss functions defined on compact spaces, the sample complexity of (realizable and agnostic) transductive learning a class H is exactly the same as the sample complexity of learning "finite projections" of H.
(2) for improper metric loss functions: there is a tight gap of 2.
(3) extensions of the above to the PAC learning model.

**Strengths:**

I think that these are nice generic results that contribute to the literature on transductive learning. Effectively, the results seem to suggest that for classes of loss functions considered in this paper, it suffices to focus attention on constructing transductive learners for finite subsets of the hypothesis class H.

The result where there is a gap of 2 (Theorem 3.8 and 3.9) is particularly interesting.

**Weaknesses:**

Below, I include some comments that may help improve the paper when addressed.

Given that the main results of the paper deals with a class H and "finite projections" of H, I think that it is important to formally define and explicitly write down what "H|_S" is. Furthermore, given that the label space Y may be infinite, H|_S may be infinite as well, in the sense that we have infinitely many projections of H onto S. I think that it is important to clarify this point, and I find it a bit misleading to refer to H|_S as a "finite projection".

The definition of transductive learning (Def 2.1, 2.2, 2.3) when unpacked boils down to defining transductive learning with respect to any n, S \subseteq X^n, and any H|_S. So, with this in mind, when looking at Theorem 3.6 and 3.8, it seems like they can be reduced to the following:
For any n, and any S \subseteq X^n,
1. H|_S is learnable with error rate epsilon.
2. For all (finite) H' \subset H|_S, H' is learnable with error rate epsilon.

What I am trying to highlight is that the results seem to be establishing equivalence between transductive learning the (potentially) infinite H|_S and transductively learning the finite subsets of H|_S. I think the authors should carefully clarify what "H is learnable in the realizable case with transductive sample function m" means. This should be included with the current Def 2.1-2.3.

Additionally, it seems that talking about error rate in transductive learning instead of sample complexity makes more sense, because as the paper defines the model (Def'n 2.1 and line 25), the adversary chooses n. So, one can just focus on smallest achievable error rate as a function of n.

------------------------------------------------------------

Based on authors' rebuttal, I raise my score to 6.

**Questions:**

See comments above.

**Limitations:**

Yes.

---

> ### Author Rebuttal · Authors · 2024-08-05
>
> We thank the reviewer for their time and attention in reviewing the paper.
>
> We note that $H|_S$ is the collection of all functions in $H$ restricted to the datapoints $S$. Indeed $H|_S$ can easily be infinite, in which case it is not a finite projection of $H$. We instead refer to the finite subsets of $H|_S$ as the finite projections of $H$, as we describe at the beginning of Section 1.1 (lines 45-48). We will clarify this point further in the next version of the paper; thank you.
>
> The reviewer’s restatements of Theorems 3.6 and 3.8 are indeed correct. We will clarify the meaning of “$H$ is learnable in the realizable case with transductive sample function $m$” alongside Definitions 2.1-2.3 to avoid confusion. (In particular, it means that $H$ has a realizable transductive learner with sample complexity at most $m$.)
>
> We agree that discussing error rates and sample complexities is equivalent, across both the transductive and PAC models. We chose to phrase our results in terms of sample complexities rather than error rates, in part arbitrarily and in part as it is roughly a convention in the community. We indeed could have equivalently phrased all results in terms of error rates throughout the paper.
>
> Thank you again for your comments on improving the exposition of the paper – they are very helpful and will be taken into account as we update the paper!

---

> > ### Comment · Reviewer_B4fj · 2024-08-08
> > **Response to rebuttal**
> >
> > Thank you for the response, I have updated my score to 6.

---

### Official Review · Reviewer_7sdt · 2024-07-12

**Soundness:** 4
**Presentation:** 4
**Contribution:** 2
**Rating:** 6
**Confidence:** 3

**Summary:**

This work studies the transductive learninng model. In this model, given  a domain $X,Y$ and $H\subset Y^X$ the adversary choses data $(x_i,y_i)$ (in the realizable setting the adversary can chose the labels after the reveal to the learner). The adversary then uniformly at random hides one data point.  The goal of the learner is to use the remaining samples to predict the hidden label. The authors first show a compactness result showing that if all the finite subsets $X'$ are learnable in the projected class, i.e., $H|X'$ with sample complexity $m$ then the same is true for the whole class. They provide a similar result for the agnostic case. Finally, the authors present a result connecting transductive learning with PAC learning.

**Strengths:**

This work provides some fundamental results for the transductive learning.  This work is well written.

**Weaknesses:**

1. The authors do not convinced me about the importance of this model. The authors should explain why this model is important, what it explains that other models does not (i.e., the classical PAC learning).
2. I do not believe that this submission is relevant to the neurips community, maybe the authors should consider submitting to COLT.
3.Furthermore, I believe that more results are needed to make this a complete submission.

**Questions:**

See 1 in weaknesses.

**Limitations:**

Yes.

---

> ### Author Rebuttal · Authors · 2024-08-05
>
> We thank the reviewer for their time and attention in reviewing the paper.
>
> > The authors do not convinced me about the importance of this model. The authors should explain why this model is important, what it explains that other models does not (i.e., the classical PAC learning).
>
> We argue that the transductive model is itself influential and insightful. Moreover, it is intimately related to the PAC model such that even those readers with an exclusive attachment to the PAC model would derive insight from our work. We tried to communicate both of these points in the paper, though acknowledge that there is room for improvement, as the review notes. We elaborate on both of these points below in order to address the reviewer’s concern.
>
> 1. The transductive approach to learning has a rich and deep history, dating to the seminal works of Vapnik and Chervonenkis ‘74 and Vapnik ‘82. Our particular model was first used by Haussler et al. ‘94 and has since been employed to derive numerous insights for PAC learning as well as other models of learning. We note that it offers several advantages relative to the PAC model, including its emphasis on labeling a single datapoint at test time, rather than emitting an entire hypothesis (i.e., the inductive vs. transductive approaches to learning). This perspective has seen renewed interest in recent years with the rise of such techniques as few-shot learning, which focuses on a model’s performance at a single test point (e.g., “Realistic Evaluation of Transductive Few-Shot Learning”, NeurIPS 2021). In fact, by focusing solely on prediction, the transductive model emphasizes improper learning by default, a notable advantage given the necessity of improper learning in settings such as multiclass classification (Daniely and Shalev-Shwartz 2014). Furthermore, the transductive model is closely related to the one-inclusion graph, a combinatorial technique that has given rise to fundamental insights across learning theory, including the first dimension to characterize multiclass learnability (the DS dimension; Brukhim et al. 2022), the first minimax optimal robust learner (via the global one-inclusion graph; Montasser et al. 2022), and the first dimension to characterize learnability for realizable regression (the \gamma-OIG dimension; Attias et al., 2023) — to name only a few examples. Lastly, the transductive model is arguably conceptually simpler than the PAC model, as it merely requires learners to attain error less than epsilon on each sample, and avoids any involvement of marginal distributions, integrability issues, confidence parameter delta, etc.
>
> 2. The transductive model bears intimate connections to the PAC model. We note that any results concerning transductive sample complexities automatically transfer to the PAC model in a black-box manner with only minor losses (lines 60-65, Lemma 3.19). Consequently, our work demonstrates an almost-exact form of compactness for sample complexities holding broadly in the PAC model (Corollary 3.20). We believe that this makes our work of central interest even to those who have an exclusive attachment to the PAC model of learning.
>
> > I do not believe that this submission is relevant to the neurips community, maybe the authors should consider submitting to COLT.
>
> We respectfully disagree with the reviewer on this point. We note that NeurIPS has published several works analyzing the transductive model and the one-inclusion graph in recent years, many of which our paper cites and builds directly upon. These include “A trichotomy for transductive online learning” (NeurIPS 2023), “Optimal learners for realizable regression: PAC learning and online learning” (NeurIPS 2023), “Adversarially Robust Learning: A Generic Minimax Optimal Learner and Characterization” (NeurIPS 2023), and “Multiclass Learnability Beyond the PAC Framework: Universal Rates and Partial Concept Classes” (NeurIPS 2022), among others.
>
> Perhaps more importantly, we believe that our work demonstrates a far-reaching structural property of sample complexities which would be of interest to many members in the NeurIPS community, both theoretically inclined and beyond. We believe that collectively, our results form an original and meaningful contribution to learning theory, as requested in NeurIPS’ call for papers.

---

> > ### Comment · Reviewer_7sdt · 2024-08-12
> >
> > I thank the authors for their response. I am raising my score to 6.

---

### Official Review · Reviewer_Rcdh · 2024-07-13

**Soundness:** 3
**Presentation:** 2
**Contribution:** 2
**Rating:** 6
**Confidence:** 3

**Summary:**

This document explores the concept of compactness in the context of transductive learning, a model closely related to the PAC model in supervised learning. The authors demonstrate that for a broad class of loss functions, a hypothesis class can be learned with a specific transductive sample complexity if and only if all its finite projections (subsets of the hypothesis class restricted to finite data sets) can be learned with the same sample complexity. This result holds for realizable and agnostic learning settings, with specific bounds provided for realizable learning with improper metric losses. The authors highlight the significance of this “exact” compactness result, as it avoids dilution by asymptotics or constants. They further connect their findings to the PAC model, revealing an almost exact form of compactness for realizable PAC learning. The paper also discusses the implications of proper versus improper learners, demonstrating a structural difference in terms of compactness. The paper's core lies in generalizing the classic marriage theorems for bipartite graphs, which provides a foundation for the compactness results.

**Strengths:**

This paper presents a compelling and rigorous analysis of compactness in the context of transductive learning. The authors contribute significantly by demonstrating an “exact” compactness result, which avoids the limitations of asymptotic or constant-based approaches. This result is particularly noteworthy for its broad applicability to a comprehensive class of loss functions and its relevance to both realizable and agnostic learning settings.
Here’s a breakdown of the paper’s strengths across different dimensions:
Originality: The paper’s originality is a key strength, stemming from its unique approach to proving compactness in transductive learning. The authors' generalization of the classic marriage theorems for bipartite graphs serves as a foundation for their key results, introducing a novel framework that establishes a precise connection between the learnability of a hypothesis class and the learnability of its finite projections, a result that has not been previously demonstrated.
Quality: The paper is of high quality, exhibiting rigorous mathematical proofs and clear exposition. The authors' careful definition of their assumptions and provision of complete proofs for all their theoretical results demonstrate a thoroughness that ensures the validity and reliability of their findings.
Clarity: The paper is well-written and easy to follow. The authors effectively introduce the concepts of transductive learning and compactness, providing clear definitions and explanations. The structure of the paper is logical, guiding the reader through the key results and their implications.
Significance: The paper’s importance lies in its contribution to our understanding of the fundamental principles of transductive learning. The “exact” compactness result provides a powerful tool for analyzing the learnability of hypothesis classes in this setting. This result can potentially impact future research in transductive learning, particularly in semi-supervised and active learning areas.
Overall, this paper presents a valuable and original contribution to the field of transductive learning. Its rigorous analysis, clear exposition, and significant implications make it a strong candidate for publication.

**Weaknesses:**

This paper presents a strong theoretical contribution, but it would benefit from a more nuanced discussion of its limitations and potential applications.

Weaknesses:

Limited Scope of Applications: While the paper establishes a powerful compactness result for transductive learning, it doesn’t delve into the practical implications of this finding. The authors could strengthen their work by exploring how this result translates to real-world scenarios. For instance, they could discuss specific transductive learning algorithms where this compactness property is particularly relevant or analyze the impact of different loss functions on the sample complexity.

Lack of Empirical Validation: The paper focuses solely on theoretical analysis. While this is valuable, it would be significantly enhanced by including empirical studies to demonstrate the practical relevance of the compactness results. Even a small-scale simulation could provide valuable insights into the behavior of transductive learners under different conditions.

Comparison to Existing Work: The paper could benefit from a more thorough comparison to existing work on compactness in learning theory. While the authors mention the PAC model, they could provide a more detailed discussion of how their results relate to existing compactness results in that framework. This would help clarify the novelty and significance of their contribution.

Discussion of Assumptions: The paper clearly states its assumptions, but it could benefit from a more in-depth discussion of their limitations. For example, the authors could explore the impact of relaxing the assumption of realizable learning or discuss the potential implications of using improper learners.

Actionable Insights:

Expand on Applications: The authors should dedicate a section to discussing potential applications of their compactness results in real-world transductive learning problems. This could involve analyzing specific algorithms, exploring the impact of different loss functions, or discussing the implications for different data distributions.

Include Empirical Studies: Even a small-scale simulation could provide valuable insights into the practical relevance of the compactness results. This would strengthen the paper’s impact and demonstrate the applicability of the theoretical findings.

Strengthen Comparison to Existing Work: The authors should provide a more detailed comparison to existing work on compactness in learning theory, particularly in the context of the PAC model. This would help clarify the novelty and significance of their contribution.
Discuss Assumption Limitations: The authors should dedicate a section to discussing the limitations of their assumptions. This could involve exploring the impact of relaxing the assumption of realizable learning or discussing the potential implications of using improper learners.

By addressing these points, the authors can significantly enhance the impact and relevance of their work.

**Questions:**

This paper presents a compelling theoretical analysis of compactness in transductive learning. However, as a reviewer, I have some questions and suggestions for the authors to consider:

1. Generalizability of Compactness Results:

Question: The paper focuses on a broad class of loss functions. Could the authors provide more concrete examples of loss functions that fall within this class and those that do not? This would help readers understand the practical implications of the results.

Suggestion: It would be beneficial to briefly discuss the results' limitations, particularly in terms of the specific loss functions that are not covered.

2. Implications of Proper vs. Improper Learners:

Question: The paper mentions a structural difference in compactness between proper and improper learners. Could the authors provide a more detailed explanation of this difference? How does it impact the practical application of the results?

A dedicated section or subsection discussing the implications of proper vs. improper learners for compactness would be very valuable.

3. Practical Applications:

Question: While the paper focuses on theoretical results, discussing potential practical applications of the compactness results would be helpful. How can these results be used to design more efficient transductive learning algorithms?

Suggestion: A brief discussion of potential applications, even if speculative, would enhance the paper’s relevance and impact.

5. Future Directions:

Question: The paper mentions a conjecture about more significant gaps between sample complexities in the agnostic case. Could the authors elaborate on this conjecture and discuss potential approaches to proving it?

Suggestion: A section on future directions, outlining potential extensions and open problems, would add value to the paper.

**Limitations:**

While the paper does not include a dedicated “Limitations” section, the authors have effectively integrated discussions of limitations throughout the paper, particularly in the introduction and conclusion.

The authors have adequately addressed the limitations of their work. They clearly state the assumptions required for their theoretical results and acknowledge that these results may not hold in more general settings. Notably, they acknowledge the potential for larger gaps between sample complexities in the agnostic case, demonstrating their awareness of the field's challenges. They also discuss the limitations of their approach in terms of its applicability to different learning settings.

However, the paper does not discuss any potential negative societal impacts of their work. This is understandable, given that the paper focuses on theoretical results in transductive learning, which is a relatively abstract field. However, it would be beneficial for the authors to briefly consider the potential applications of their work and any potential negative societal impacts that might arise.

---

### Decision · Program_Chairs · 2024-09-25

**Decision:**

Accept (poster)

**Comment:**

The authors show a "compactness" result for transductive learning for a range of settings and loss functions. Roughly speaking, they show the sample complexity of transductive learning is m iff it is m for all of its "finite projections" (which means to characterize sample complexity it is enough to look at finite subsets of the class). "Compactness" is a rather mathematical concept and is a bit far from usual NeurIPS papers. However, the reviewers found the techniques and results interesting and relevant to the theory community. As such, I also recommend acceptance.

Reviewer Rcdh's comment were not used in the decision making by the AC.